



# Airborne measurements of turbulence and cloud microphysics during PaCE 2022 using the Advanced Max Planck CloudKite Instrument (MPCK+)

Oliver Schlenczek[1], Freja Nordsiek[1,2], Claudia E. Brunner[1], Venecia Chávez-Medina[1], Birte Thiede[1], Eberhard Bodenschatz[1,3,4], and Gholamhossein Bagheri[1]

[1]Max Planck Institute for Dynamics and Self-Organization (MPI-DS), Am Faßberg 17, 37077 Göttingen, Germany
[2]Gesellschaft für wissenschaftliche Datenverarbeitung mbH Göttingen (GWDG), Burckhardtweg 4, 37077 Göttingen, Germany
[3]Institute for Dynamics of Complex Systems, Georg August University of Göttingen, Friedrich-Hund-Platz 1, 37077 Göttingen, Germany
[4]Laboratory of Atomic and Solid State Physics and Sibley School of Mechanical and Aerospace Engineering, Cornell University, 130 Upton Hall, Ithaca NY 14853, USA

**Correspondence:** Gholamhossein Bagheri (gholamhossein.bagheri@ds.mpg.de)

**Abstract.**

This paper presents the data set collected with the Advanced Max Planck CloudKite instrument (MPCK+) during the Pallas Cloud Experiment (PaCE), conducted at Pallas, Finland, between September 19 and September 26, 2022. The data set includes measurements of turbulence, wind shear, and cloud microphysics in the planetary boundary layer between 0 and 1200 m above ground with flight durations between 1.5 and 3 hours. This data set is intended for researchers studying microphysics of boundary layer clouds in the Arctic at temperatures above freezing level, providing high-resolution airborne measurements of meteorological variables together with cloud droplet concentration and size distribution data. In particular the high spatial resolution of less than 10 m between two cloud droplet size distribution samples, the low altitude and the long flight time allow unprecedented insights into stratocumulus and cumulus clouds not accessible by research airplanes. The aim of this paper is to describe the data collection process, the structure of the data set, and guidelines for potential users.

## 1 Introduction

The interplay of clouds and turbulence, in particular the formation of warm rain, is subject to ongoing research efforts world wide in order to improve numerical climate and weather predictions. Due to the high degree of turbulence in a typical cumulus cloud, measurements in the laboratory are generally not suitable to yield a deeper understanding of the droplet dynamics and microphysics under real atmospheric conditions. Our approach is to perform in-situ measurements using a tethered balloon system. As opposed to research aircraft, the air speed measured by a tethered balloon instrument is much lower, which allows for much higher spatial and temporal resolution. While airplanes fly at 50 to 150 $\mathrm{m\,s^{-1}}$, the air speed of a tethered balloon system is that of the wind itself, on the order of 5 to 10 $\mathrm{m\,s^{-1}}$, thus increasing the spatial resolution by a factor 5 to 30. Another





advantage of a tethered balloon system is the possibility to fly under conditions which are too risky for pilots of air planes, e.g.
at low altitude over complex terrain.

Our approach combines various sensors for different atmospheric and cloud variables into one scientific payload: the Advanced Max Planck CloudKite instrument (MPCK[+]). The MPCK[+]is the first instrument developed in-house for the Max Planck CloudKite (MPCK) platform. The main unit of the MPCK[+]is the combined Particle Imaging Velocimetry (PIV) and inline holography unit. The PIV unit can measure particle velocity in stream-wise and vertical direction, and its sample volume overlaps partly with the holographic sample volume. The inline holography unit records 75 images per second, the PIV unit records 15 image pairs per second. Every fifth hologram has a corresponding PIV image pair at exactly the same time. This unique design allows in-situ inter-instrument calibration as cloud droplets are detected by both instruments in the overlapping volume. The low air speed together with the high image acquisition rate yields inter-hologram distances on the order of 5 to 6 centimetres. Besides this complex setup for detailed measurement of cloud droplets in 2D and 3D, the MPCK[+]has a commercial Fast Cloud Droplet Probe (FCDP) for additional 1D measurements of cloud droplets. One feature of the MPCK[+]control software is that particles detected by the FCDP above a given size and concentration threshold can be used to automatically trigger the holography and PIV unit. Data acquired by the two imaging units are not part of the published data set, but the flight table provides information about availability of these data.

Besides the data recorded by the imaging units, the MPCK[+]measures basic atmospheric variables like static air pressure, air temperature, relative humidity, air speed relative to the MPCK[+], particle number concentration, particle diameter, orientation and velocity of the MPCK[+]itself. These data are measured by multiple sensors to allow for redundancy of essential variables and in-situ instrument intercomparison. Our measurements provide valuable insights into clouds on spatial scales smaller than 10 m. Also, the thermodynamic and dynamic properties of the atmospheric boundary layer can be studied using these data, for example the role of wind shear and convection in different meteorological conditions. In addition to these basic variables, the MPCK[+]measures turbulent fluctuations of the wind velocity using a thermal anemometry probe. The details of the MPCK[+]are summarised in another paper (Bagheri et al., 2025a).

## 2 Methods and Data Collection

### 2.1 The PaCE 2022 field campaign

The aim of the Pallas Cloud Experiment (PaCE) was the characterisation of aerosols and clouds in the vertical column at high resolution around the Pallas-Sodankylä Global Atmospheric Watch (GAW) Sammaltunturi station in Finnish Lapland, approximately 160 km north of the Arctic Circle (Doulgeris et al., 2022; Brus et al., 2025; Gratzl et al., 2025). The field campaign took place from September 15 to December 15, 2022 and was organised by the Finnish Meteorological Institute (FMI). Besides the Max Planck Institute for Dynamics and Self-Organization (MPI-DS) and the FMI, the Swiss Federal Institute of Technology Lausanne, the Karlsruhe Institute of Technology, the University of Hertfordshire, and the Vienna University of Technology participated in this campaign. The measurement were taken via fixed-wing and multi-copter drones, tethered balloon systems, mountain-top stations, ground stations and remote sensing instruments. A general overview within





this Special Issue is given in Brus et al. (2025). In this paper we present data obtained with the MPCK[+]via in-situ measurements aboard the tethered balloon system Max Planck CloudKite (MPCK) between September 19 and September 26, 2022.

The site from which the balloons with the instruments were launched is within the Pallas-Yllästunturi National Park in Finnish Lapland at the western shoreline of lake Pallasjärvi, approximately 280 m above mean sea level (MSL) [1]. Northwest of the site is a range of hills with up to 640 m MSL. The site itself is grassland, which is surrounded by boreal forest. The two main wind directions are south-west (towards the lake) and east (from the lake). The lake has a strong influence on temperature and wind in the lower atmospheric boundary layer (ABL) around the site, in particular during early summer (colder than the surrounding area) and late autumn (warmer than the surrounding area). More details about the site and the campaign can be found in Chavez-Medina et al. (2025) and Brus et al. (2025).

## 2.2 The Max Planck CloudKite (MPCK) platform

In this section, we introduce the Max Planck CloudKite platform for airborne atmospheric measurements, abbreviated MPCK platform. It consists of one or more helium-filled tethered Helikites (type Desert Star Helikite by Allsopp Helikites Ltd.) and the accessories needed for flight operation (ground anchor, pulley, motor winch with main tether, etc.). The MPCK platform can carry different scientific payloads, such as the MPCK[+](Stevens et al., 2021) or the WinDarts (Chavez-Medina et al., 2025).

A Helikite is a balloon-kite combination whose main advantages over an ordinary balloon are the keel and sail underneath the bubble, which stabilise the aerostat in a 45 °angle in windy conditions. The flight altitude, typically up to 2 km, is controlled by a motorised winch reeling in/out the main tether. By changing the tether length, the flight altitude is regulated. The maximum recommended wind speed for use is $25 \, \mathrm{m\,s^{-1}}$. During PaCE 2022, the wind speed was always below $20 \, \mathrm{m\,s^{-1}}$. A photo from a test site next to the MPI-DS shows the MPCK platform with one 250 $\mathrm{m^3}$ Helikite (figure 1). During most of the PaCE2022 flights, we operated a tandem configuration with one 250 $\mathrm{m^3}$ Helikite as the lower balloon and one 34 $\mathrm{m^3}$ Helikite as the higher balloon. Flight strategies with the MPCK platform include constant altitude flights, vertical profiles, and staircase flights. The combination of the Helikites' flight properties and the winch height control makes in-situ measurements highly adaptable. The static and dynamic lift generated by the Helikites was sufficient to reach altitudes of more than 1200 m above ground with 100 kg payload.

While the MPCK platform has been used for flying different scientific payloads during PaCE 2022, we focus on the MPCK[+]instrument in this paper. A close view of the MPCK[+]installation on the main tether is shown in figure 2. This type of installation was chosen in order to minimise the influence of the balloons on the dynamics measured by the MPCK[+]instrument. A particular challenge is the measurement of atmospheric variables under ambient conditions, i.e. the measurements not being influenced by the instrument deployment or instrument geometry. Here, the MPCK[+]is hanging on the main tether at least 50 to 100 m below the 250 $\mathrm{m^3}$ Helikite, which is the equivalent vertical distance of 5 to 10 balloon diameters. This mounting option is called tether-mount in Bagheri et al. (2025a). One side effect of this setup is the non-linear relationship between line length and instrument altitude due to the catenary, which means that the gain of altitude decreases with increasing line length.

---

[1]For location details, visit https://en.ilmatieteenlaitos.fi/pallas-atmosphere-ecosystem-supersite

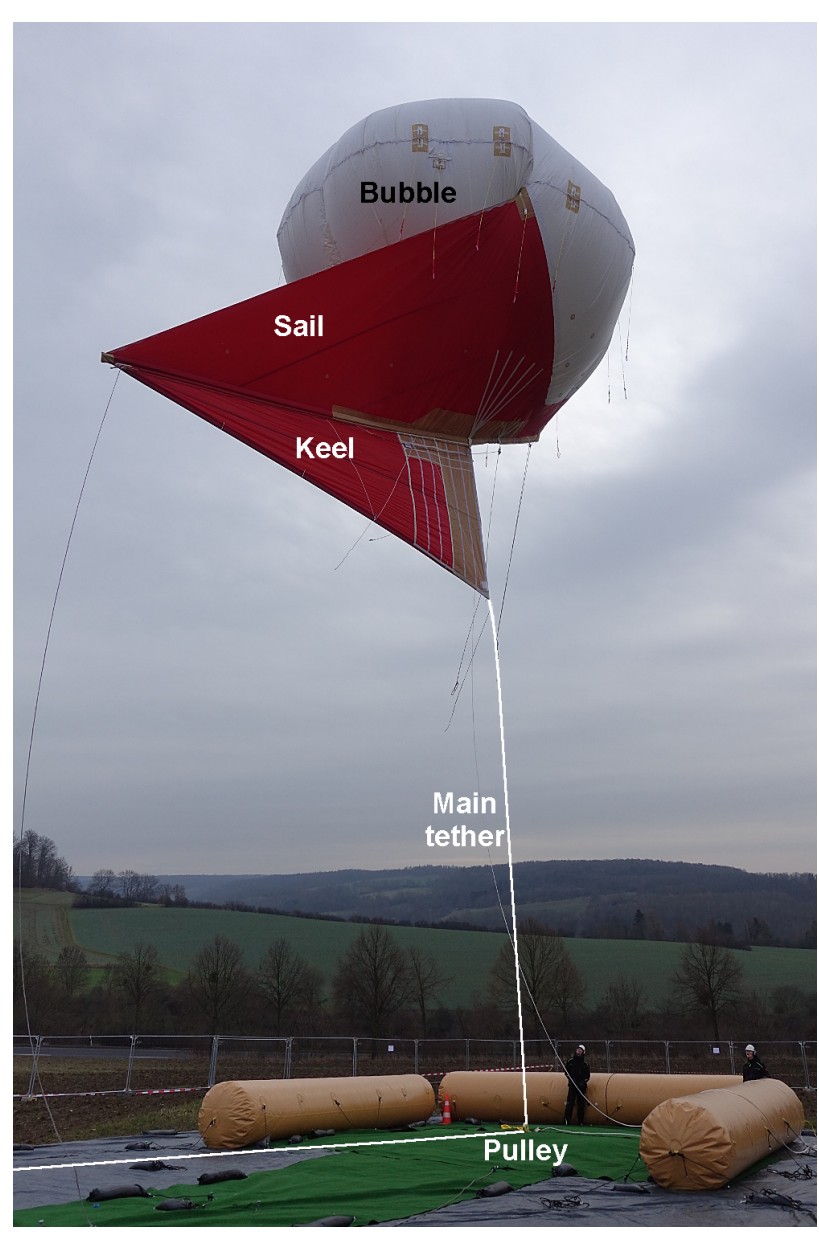

**Figure 1.** A 250 m³ Helikite during a test flight.



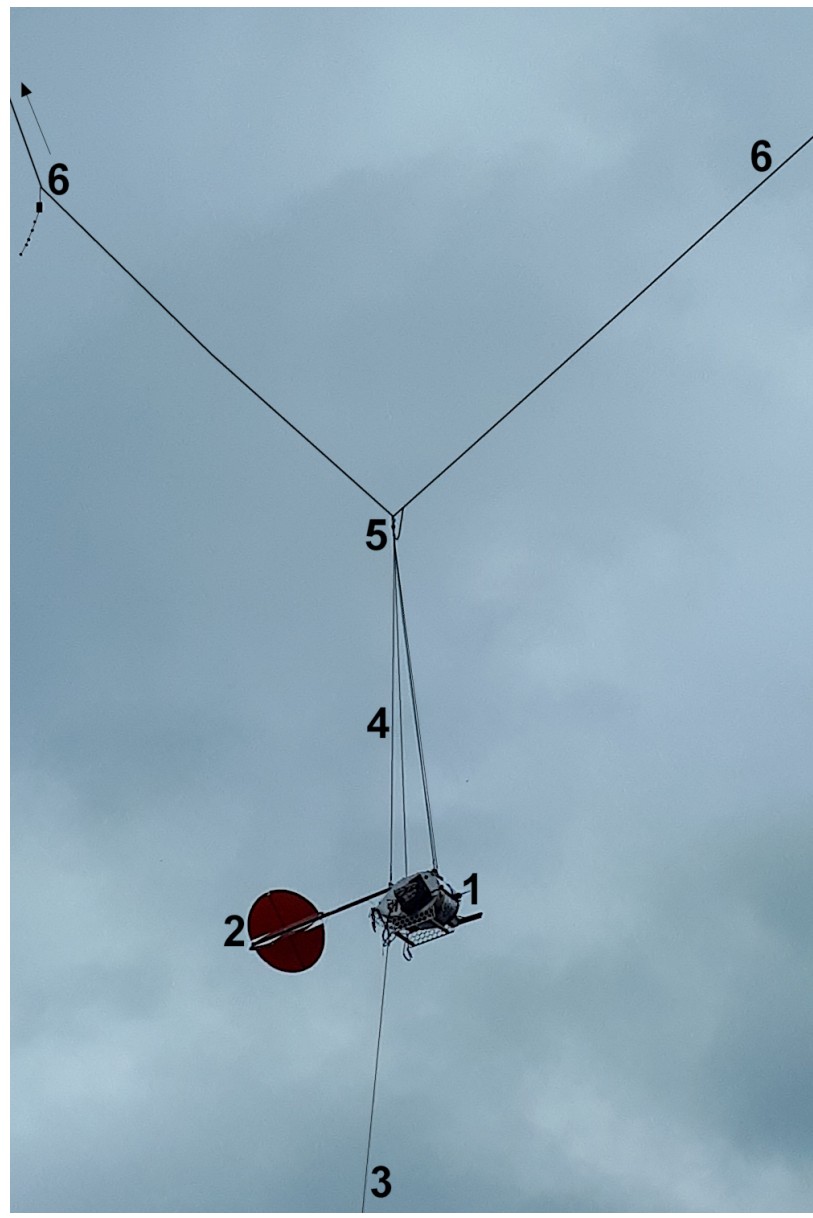

**Figure 2.** MPCK$^+$instrument (1) with stabiliser fin (2) hanging on four line segments (4) connected to the main tether (6) by a knot (5). It has a handling line (3) hanging below for guiding it during take-off and landing. The arrow next to the main tether (6) points towards the two Helikites.





| Sensor | Device | Manufacturer | Quantity | Acq. rate [Hz] |
|---|---|---|---|---|
| Ellipse D | sbg | SBG Systems | Position, velocity, acceleration, angular velocity, magnetic field, platform orientation | 200 |
| PSS8 | pss8 | Simtec | Temperature, absolute pressure, air speed | 100 |
| HMP 7 | hmp7 | Vaisala | Temperature and relative humidity | 1 |
| AM2315 | rht | Aosong | Temperature and relative humidity | 0.5 |
| 54T42 miniCTA & Labjack DAQ | labjack | Dantec Labjack | Air speed | 65536 |
| NEO-M8U backup GNSS | powerinterlock | ublox | Position, velocity, acceleration, angular velocity, platform orientation | 1 |
| FCDP | fcdp | SPEC | Particle number concentration (2-50 μm), particle diameter | 1 |
| PIV/PTV | piv | in-house | 2D particle velocity, diameter | 15 |
| Inline holography | holo | in-house | Particle shape, diameter and concentration | 75 |

**Table 1.** Instrumentation of the MPCK[+]instrument.

## 2.3 Instrumentation: the MPCK[+]

The MPCK[+]consists of nine devices to measure different platform and atmospheric properties. A summary of the instrument specifications is given in table 1. The core unit consists of a one-dimensional (1D) pitot static system (Model type PSS8, Simtec AG) for measurement of air speed, barometric altitude and temperature, an inertial navigation system (Model Ellipse D, SBG Systems) for measurement of the platform position, orientation, velocity, angular velocity, acceleration and angular acceleration, a humidity and temperature sensor unit (labeled "rht" in table 1) with an AM2315 sensor (Guangzhou Aosong Electronic

Co., Ltd.) and a nearby HMP7 sensor (Vaisala), a backup global navigation satellite system (GNSS), model NEO-M8U (uBlox), and a thermal anemometry unit, which comprises a 54T42 mini-CTA (Dantec Dynamics) and a T7 Data Acquisition unit (Lab-Jack Corporation). Particle size and concentration in 1D are measured by a Fast Cloud Droplet Probe (FCDP) from Stratton Park Engineering Company (SPEC Inc.). For measurement of 3D particle position, size and shape, an in-house- built digital inline holography unit is used. Particle velocity in the stream-wise-vertical plane is measured with an in-house-built Particle

Imaging Velocimetry (PIV) unit.

     Besides the MPCK[+]Instrument, additional scientific instruments were operating simultaneously: the WinDarts (see Chavez-Medina et al., 2025), and the FishBox, which was developed by the Finnish Meteorological Institute (FMI) team and comprises





several aerosol particle size spectrometers, a GPS and meteorological instrumentation (see Brus et al., 2025, for details). Along with the MPCK platform and its scientific payloads, we operated a ground weather station next to the launch site. This
weather station was equipped with a WS500 UMB combined weather monitor (Lufft) (for static air pressure, temperature, relative humidity and 2D wind), a uSonic Class A-MP 3D sonic anemometer (for high resolution 3D wind and temperature), a LI-7500DS optical trace gas analyser (for high resolution data of static air pressure, temperature, water vapour and $CO_2$ concentration), and a CS 110 field mill (for the vertical component of the atmospheric electric field). In addition to these instruments, we operated a LD 250 lightning detector and a pingStation Mode S and ADS-B receiver for safety reasons
(uAvionix), i.e. to monitor the risk of thunderstorms and lightning and to monitor the airspace. More details about the ground monitoring and a schematic of the setup are presented in Chavez-Medina et al. (2025).

## 2.4 Flight overview

Between September 19 and September 26, 2022, we performed eight flights with the MPCK+with a total flight time of 19 hours and 37 minutes. The longest flight took 3 hours and 12 minutes, the shortest flight was 1 hour and 32 minutes long. While the
110 lowest flight had a maximum altitude of 240 m above ground, the highest flight was at maximum 1210 m above ground. A summary of all the flights is given in table 2. Figure 3 shows the barometric altitude time series records for flights 20220919.1236, 20220921.1721, 20220923.0711, 20220923.0914, 20220923.1622, 20220925.0839, 20220926.1209 and 20220926.1712. The first six flights were done with the tandem configuration whereas the flights on 26 September 2022 were done with just the 250 m$^3$ Helikite. Flights 20220923.0711, 20220925.0839 and 20220926.1209 had at least one WinDart measuring simultane-
115 ously with the MPCK+at different altitude. These data are published in Chavez-Medina et al. (2025).

During Flight 20220919.1236 there was an issue with the power supply of the devices, which is visible by the data gaps of the altitude time series. This flight had broken cumulus and stratocumulus with no precipitation and northerly wind. The issue with the power supply was fixed and did not occur during any of the other flights. Flight 20220921.1721 had southwesterly wind and broken altocumulus and altostratus. The cloud base was not reached by the MPCK+in this flight. This
flight was the one with the strongest measured temperature inversion as shown in Section 3.5. Flight 20220923.0711 had broken cumulus and stratocumulus with southerly wind and the MPCK+stayed below cloud base again. The cloud and wind conditions did not change much until the end of Flight 20220923.0914, in which the MPCK+recorded some brief cloud events. Flight 20220923.1622 had southerly wind and few cumulus clouds, the MPCK+did not reach the cloud layer in this flight. Flight 20220925.0839 had broken to overcast stratocumulus and southerly wind, the MPCK+recorded a few brief cloud events.
During this flight, there were some issues with the measured air speed. Flight 20220926.1209 had low overcast cumulus with drizzle and south-easterly wind and the MPCK+stayed in the cloud during most of the flight time. This flight has the largest data set of good quality holograms, the holographic data cover almost 9 minutes of flight time. An example is shown in Section 3.6. However, there was again an issue with the measured air speed, which happened around 13:03 UTC. We believe there was water inside the tubing which connects the pitot tube and the pressure transducer. Flight 20220926.1712 had low overcast
cumulus with rain and south-easterly wind and cloud conditions similar to the previous flight. Again, the MPCK+stayed inside





the cloud during most of the flight time. This time, the velocity measurement is valid until 19:00 UTC but the number of high quality holograms covers only one minute of the time series.

## 3 Data

### 3.1 File Structure

The data are provided in NetCDF format, following a common file naming structure. This structure is as follows:
**MPDS.MPCKplus.b1.yyyymmdd.hhmm.nc**, where:

- **MPDS** is the institute identifier (Max Planck Institute for Dynamics and Self-Organization).

- **MPCKplus** is the instrument identifier (Advanced Max Planck CloudKite).

- **b1** indicates the data file processing level, with quality control (QC) checks applied; missing data points or those with
140 bad values can be eliminated by forcing the values to be smaller than $9 \cdot 10^3 6$.

- **yyyymmdd.hhmm** is the flight identifier (flight ID).

- **yyyymmdd** denotes the file date (UTC) in year, month, day format.

- **hhmm** represents the file start time (UTC) in hours and minutes format.

- **.nc** is the file extension for the NetCDF.

Each data file contains 50 attributes (institution, contact information, standard name vocabulary, campaign time, campaign location, flight number and many more) and one group, which is labeled "Level_1". This group contains 10 subgroups and 51 attributes. Each subgroup represents one particular measurement device. The device names are fcdp (Fast Cloud Droplet Probe), hmp7 (temperature and humidity sensor), powerinterlock (backup GNSS and some housekeeping data), pss8 (pitot static system), rht (the other temperature and humidity sensor), and sbg (inertial navigation system). An overview of the data
structure for the different devices is given in table 3. Each group and data set contains attributes which describe the type of data, the calibration factors applied, and the physical units.

Not all devices mentioned in table 2 are included in the data published in this paper. The additional groups, which are excluded from the published data, are labjack, holo, piv and serialcameras. Access can be granted by the MPI-DS upon reasonable request.

Within the comprehensive list of variables contained in the published netCDF files, there is a subset of core variables that have been used to do the analyses and generate the figures presented in this paper. Table 4 lists and describes the most important variables, which are usually enough for most users. For generating these variables, the calibration polynomials either from the manufacturer or an in-house calibration have been applied to the raw data. Especially for the true air velocity values, which



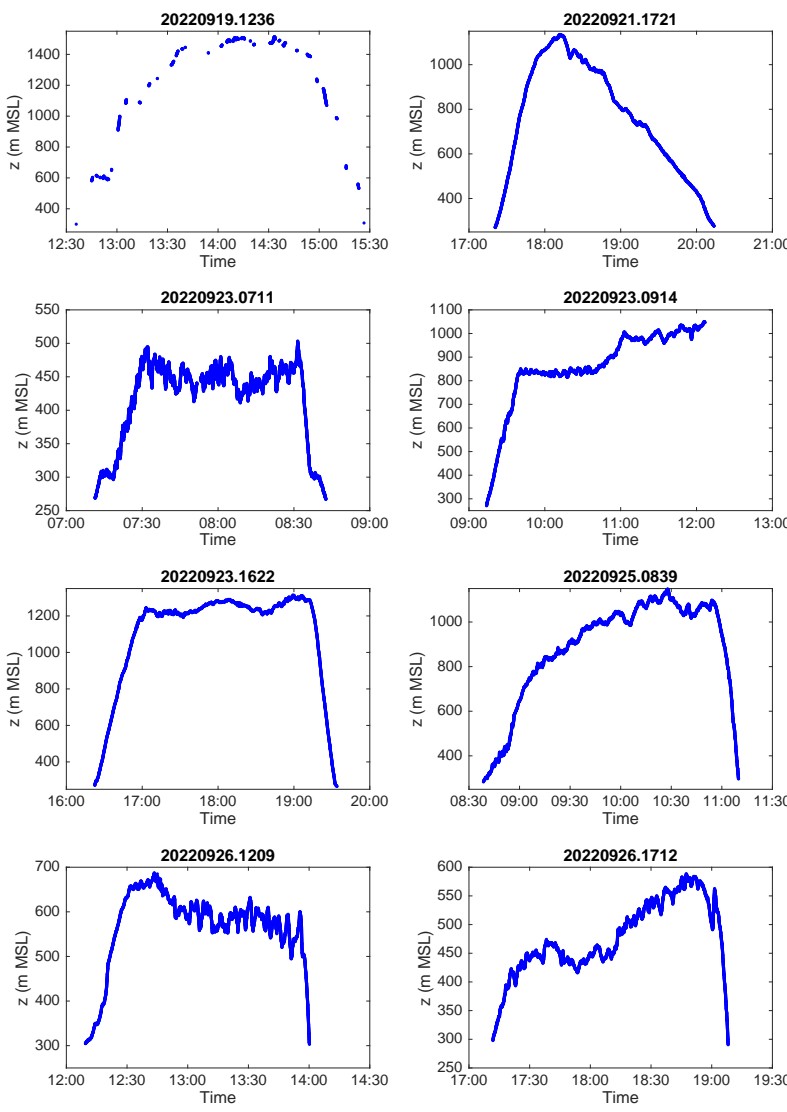

**Figure 3.** Time series of barometric altitude measured with the pss8 device aboard the MPCK⁺instrument, the flight numbers are given in each panel title. Flight 20220919.1236 (top row) had issues with the power supply, resulting in visible gaps of the data record.



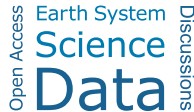

| Flight ID | Instrument | Start of recording time (UTC) | Take-off time (UTC) | End of recording time (UTC) | Landing time (UTC) | Total duration of flight | Data available | Max. altitude [m] |
|---|---|---|---|---|---|---|---|---|
| 20220919.1236 | MPCK+ | 12:32:16 | 12:36 | 15:35:44 | 15:26 | 2h 50m | pss8, sbg, hmp7, rht, powerinterlock, webcams | 1210 |
| 20220921.1721 | MPCK+ | 16:54:00 | 17:21 | 20:34:27 | 20:14 | 2h 53m | pss8, sbg, hmp7, rht, powerinterlock, webcams, labjack, fcdp | 860 |
| 20220923.0711 | MPCK+ | 06:54:13 | 07:11 | 08:59:59 | 08:43 | 1h 32m | pss8, sbg, hmp7, rht, powerinterlock, webcams | 240 |
| 20220923.0914 | MPCK+ | 09:00:00 | 09:14 | 12:06:36 | 12:14 | 3h 00m | pss8, sbg, hmp7, rht, powerinterlock, webcams, labjack, fcdp, holo, piv | 780 |
| 20220923.1622 | MPCK+ | 15:52:04 | 16:22 | 19:58:00 | 19:34 | 3h 12m | pss8, sbg, hmp7, rht, powerinterlock, webcams, labjack, fcdp | 1050 |
| 20220925.0839 | MPCK+ | 08:32:02 | 08:39 | 11:47:36 | 11:10 | 2h 31m | pss8, sbg, hmp7, rht, powerinterlock, webcams, labjack, fcdp, piv | 850 |
| 20220926.1209 | MPCK+ | 11:46:07 | 12:09 | 14:11:49 | 14:00 | 1h 51m | pss8, sbg, hmp7, rht, powerinterlock, webcams, labjack, fcdp, holo, piv | 390 |
| 20220926.1712 | MPCK+ | 17:00:12 | 17:12 | 19:19:13 | 19:08 | 1h 56m | pss8, sbg, hmp7, rht, powerinterlock, webcams, labjack, fcdp, holo, piv | 300 |

**Table 2.** Overview of flights with the MPCK+ during the PaCE field campaign. The times for take-off and landing are approximations derived from altitude data. Altitude is given in m above ground. Our naming convention is **MPDS.MPCKplus.b1.yyyymmdd.hhmm.nc**, where: **MPDS** is the institute identifier, **MPCKplus** is the instrument identifier, **b1** indicates the data file processing level, **yyyymmdd** denotes the file date (UTC) in year, month, day format, and **hhmm** represents the take-off time (UTC) in hours and minutes format. The cumulative flight duration across all flights totals 19 hours and 37 minutes.

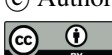



| Device | Group | Subgroup | Data description |
|---|---|---|---|
| fcdp | fcdp | basic_histogram_status | data from FCDP acquisition program |
| | | better_histograms | data from the FCDP SD card |
| | | initial_configuration | Configuration from FCDP acquisition program |
| | | particle_by_particle | Time and diameter of every measured particle |
| | | waveforms | FCDP raw data of each measured particle |
| hmp7 | hmp7 | - | Temperature and humidity data |
| powerinterlock | powerinterlock | gnss_pps | Received PPS signal from GNSS |
| | | status | Microcontroller info |
| | | ubx_esf_ins | Data messages from ublox ESF INS |
| | | ubx_esf_status | Data messages from ublox ESF status |
| | | ubx_nav_att | Data messages from ublox NAV ATT |
| | | ubx_nav_pvt | Data messages from ublox NAV PVT |
| | | ubx_nav_sol | Data messages from ublox NAV SOL |
| pss8 | pss8 | - | Data from Pitot Static System |
| rht | rht | rht | Temperature and humidity data |
| sbg | sbg | euler | Euler angles |
| | | filtered_nav | Filtered position and velocity |
| | | filtered_quaternion | Filtered orientation in quaternion format |
| | | gps_hdt | Pitch and yaw angle from both antennas |
| | | gps_pos | GNSS based position |
| | | gps_vel | GNSS based velocity |
| | | imu | Raw IMU data |
| | | mag | Magnetometer and accelerometer data |
| | | utc_time | GNSS based time reference |

**Table 3.** Structure of data groups and subgroups for each device.

| Device | Group | Data set | Description |
|--------|-------|----------|-------------|
| fcdp | better_histograms | time | Time in UTC seconds since 01 January 1970 |
| fcdp | better_histograms | particle_bin_counts_2darray | Particle counts for each size bin per second |
| fcdp | better_histograms | particle_diameter_bin | Bin mid points of particle size bins in the bin counts 2d array in m |
| fcdp | particle_by_particle | time | Time in UTC seconds since 01 January 1970 |
| fcdp | particle_by_particle | particle_diameter | Particle diameter in m |
| hmp7 | | time | Time in UTC seconds since 01 January 1970 |
| hmp7 | | air_temperature | Air temperature in K with calibration factor applied |
| hmp7 | | relative_humidity | Relative humidity (value of 1 for water saturated conditions) with calibration factor applied |
| powerinterlock | ubx_nav_pvt | time | Time in UTC seconds since 01 January 1970 |
| powerinterlock | ubx_nav_pvt | alt | GNSS altitude in m MSL |
| powerinterlock | ubx_nav_pvt | lat | Geographic latitude in ° |
| powerinterlock | ubx_nav_pvt | lon | Geographic longitude in ° |
| powerinterlock | ubx_nav_att | time | Time in UTC seconds since 01 January 1970 |
| powerinterlock | ubx_nav_att | platform_pitch_fore_up | Pitch angle in ° |
| powerinterlock | ubx_nav_att | platform_roll_starboard_down | Roll angle in ° |
| powerinterlock | ubx_nav_att | platform_yaw_fore_starboard | Yaw angle in ° |
| pss8 | | time | Time in UTC seconds since 01 January 1970 |
| pss8 | | instrument_calibrated_speed_wrt_air | Air speed in m/s with manufacturer calibration factor applied |
| pss8 | | barometric_altitude | Altitude in m above mean sea level (MSL) derived from air pressure |
| pss8 | | air_temperature | Air temperature in K with calibration factor applied |
| pss8 | | air_pressure | Static air pressure in Pa |
| rht | | time | Time in UTC seconds since 01 January 1970 |
| rht | | air_temperature_2 | Air temperature in K with calibration factor applied |
| rht | | relative_humidity_2 | Relative humidity (value of 1 for water saturated conditions) with calibration factor applied |
| sbg | euler | time | Time in UTC seconds since 01 January 1970 |
| sbg | euler | platform_pitch_fore_up | Pitch angle in ° |
| sbg | euler | platform_roll_starboard_down | Roll angle in ° |
| sbg | euler | platform_yaw_fore_starboard | Yaw angle in ° |
| sbg | filtered_nav | time | Time in UTC seconds since 01 January 1970 |
| sbg | filtered_nav | alt | GNSS altitude in m MSL |
| sbg | filtered_nav | lat | Geographic latitude in ° |
| sbg | filtered_nav | lon | Geographic longitude in ° |

**Table 4.** Core variables in the published netCDF files discussed in this paper





are also required for the calculation of the droplet concentration, corrections of the air velocity values are necessary due to the geometry of the MPCK$^+$, more on this are provided on the section 4.

At this point it needs to be mentioned that there are two more correction factors, which have not been applied to the data. These correction factors result from tests with the MPCK$^+$ in our wind tunnel to determine the disparity between undisturbed and measured air speed at the pitot tube and other locations, e.g. at the probing volume of the FCDP for calculation of droplet concentration. To correct the calibrated air speed from pss8 to ambient air speed (at infinity), the data need to be divided by 0.647. At the sample volume of the FCDP, the actual air speed is the measured air speed multiplied by 1.09/0.647. These correction factors are included in the sample Python code linked to this document to ensure that users are given clear instructions on how to calculate the concentration and apply various corrections, and to facilitate the use of data.

### 3.2 Level 1 data generation

Different devices and microcontrollers acquire the data of the various instruments of the MPCK$^+$. This requires time synchronisation of all instruments against one master clock. The synchronisation is the very first step in the data processing from raw data to Level 1 processed data. After synchronisation, the data are checked for physical plausibility to detect obvious issues due to a broken sensor, a loose connection or other reasons. In the next step, the calibration polynomials are applied to the raw data; they are either from a calibration certificate of the manufacturer or based on an in-house calibration. Data sets of a missing sensor have all values above $9.9 \cdot 10^3 6$. Anything larger than $2^6 4$ is considered invalid data or missing data.

### 3.3 Temperature and relative humidity data

To clarify the accuracy of humidity measurements, we chose a flight which had mostly clouds at the instrument altitude. For this reason we selected flights 20220925.0839 and 20220926.1209 as examples. Starting with Flight 20220925.0839 as shown in figure 4, we see a strong positive deviation of the pss8 temperature data compared to the rht and hmp7 units of about +3.5 K. Furthermore, the hmp7 dew point and temperature data suggest high supersaturation during the entire period whereas the FCDP detected particles only in a few short events during the flight. As the deviation between the rht and hmp7 units in temperature is typically smaller than 200 mK, we trust both sensors and also the arithmetic mean of both in terms of temperature. The dew point, however, only matches with the FCDP data if we take the arithmetic mean of both instruments.

A different example is shown in figure 5 for Flight 20220926.1209. Here, the instrument was flying in clouds for most of the entire flight time. Again, we see very little deviation in the temperature data between rht and hmp7. The strong deviation in the dew point temperature is about the same as in the previous example, and also here, the mean dewpoint is more trustworthy than any of the individual dew point data. Based on the results shown in figures 4 and 5, we decided to show only mean temperature, mean dew point, pss8 temperature and ground station temperature in the following graphs instead of plotting data for each individual sensor.

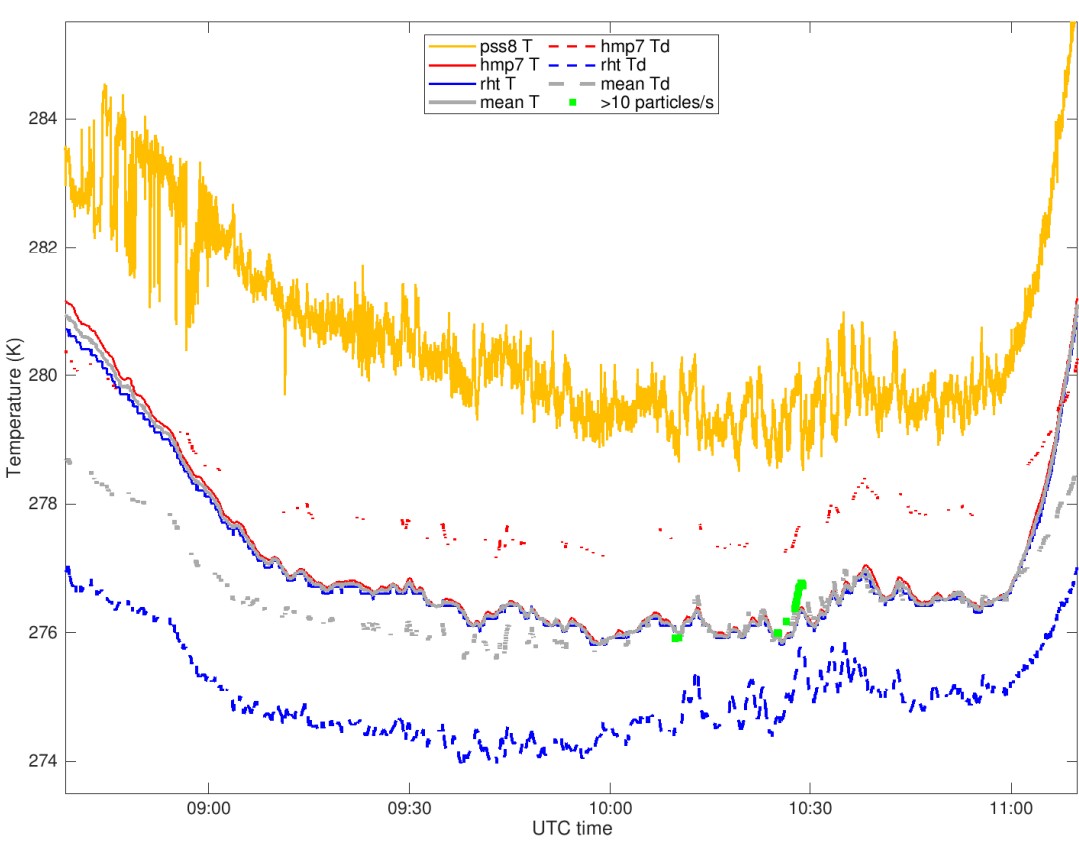

**Figure 4.** Temperature (solid lines) and dew point temperature (dashed lines) measured by / calculated from different MPCK$^+$ sensors for Flight 20220925.0839 as a function of time. Shown are temperature data from the pss8 system (orange), temperature data from the hmp7 sensor (red), temperature data from the rht sensor (blue), dew point data from the hmp7 sensor (dashed red), dew point data from the rht sensor (dashed blue), calculated mean temperature (solid gray), and calculated mean dew point (dashed gray). In addition, the green squares indicate times when the FCDP measured 10 or more particles larger than 10 μm, which indicates the presence of a cloud. Mean values are the arithmetic mean values from the rht sensor and the hmp7 sensor with equal weights for both sensors.

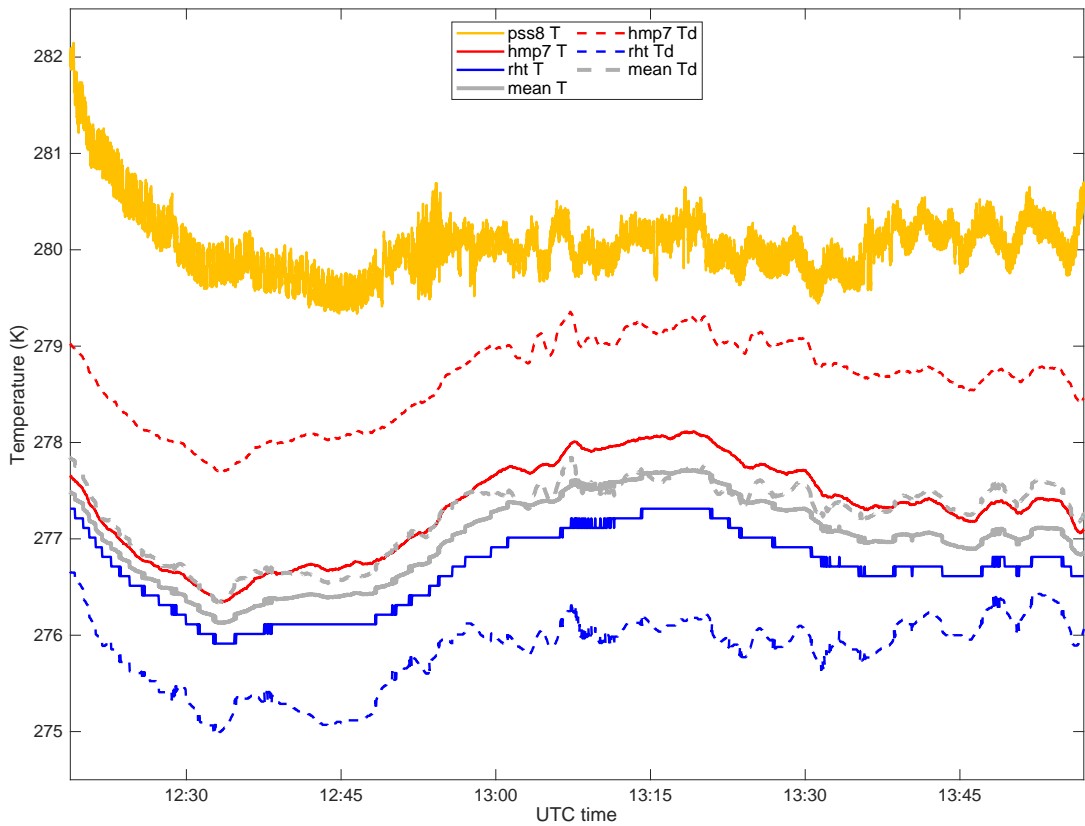

**Figure 5.** As in figure 4 for Flight 20220926.1209. Here, the entire flight was within clouds.

### 3.4 In-situ data with and without clouds measured

In this section we present example time series data from the different sensors of MPCK⁺during two selected flights. One of them was without clouds at the instrument altitude, during the other one the instrument was inside the cloud throughout most of the flight time.

One example of data from a flight without clouds and precipitation (Flight 20220921.1721) is presented in figure 6. The barometric altitude and GPS altitude in the top panel have little deviation before 18:10 and after 18:50 UTC. In between, there
is a deviation of 15 to 20 m between the sbg altitude and ublox altitude, which is due to the fact that the sbg has a dual band GNSS. The second row of figure 6 shows the pitch and roll angle of the MPCK⁺. At lower air speed (around $10\,\mathrm{m\,s^{-1}}$ visible in the third row of figure 6), the instrument orientation is almost ideal (pitch angle close to zero, roll angle slightly positive). With stronger wind, the pitch angle becomes negative, but no more than 10 degrees. At these angles, the velocity measurement is not





impaired by non-ideal platform orientation. In comparison to the wind speed at ground level (blue line in figure 6 third row),
the upper air wind speed is between 8 and 15 m s$^{-1}$ higher. The second to last panel in figure 6 shows mean temperature and
mean dew point as defined in the previous paragraph along with the temperature measured by the pss8 sensor and the ground
station temperature. At about 200 m above ground, the mean temperature is still about 0.5 K higher than the temperature
measured by the ground station, which suggests an influence by the heat capacity of the MPCK$^+$. After take-off, it took about
15 minutes for the sensors to equilibrate to ambient temperature. The humidity values reflected by the dew point temperature
(thick blue line) show the exact same trend but are separated by 4 K in dew point temperature. The temperature sensor of the
pitot static system (pss8) is positively biased with respect to the mean temperature by approximately 2 K. The bottom panel
in figure 6 shows the particle size and concentration measured by the FCDP as a function of time. There are particle counts at
low concentration between 3 and 15 μm particle diameter, which most likely correspond to a few big aerosol particles. These
particles were present below 800 m MSL.

In contrast to Flight 20220921.1721, Flight 20220926.1209 was mostly inside clouds as shown in figure 7. This is indicated
by much higher particle concentration in the FCDP data. The cloud base was reached only about 70 m above ground during
this flight, and particle data from the FCDP indicate a bimodal cloud droplet size distribution (CDSD) throughout the entire
cloud portion that was sampled by the MPCK$^+$. The primary mode around 5 μm corresponds to the nucleation mode right after
activation of cloud condensation nuclei (CCN). The secondary mode around 20-25 μm particle diameter is the accumulation
mode due to water vapour diffusion. There is also a strong signal of particles larger than 40 μm diameter, which correspond
to those droplets that turn into drizzle by collision and coalescence. The analysis of the entire CDSD of this cloud, including
droplets larger than 50 μm, is subject to a follow-up study with additional data from the inline holography and PIV units. The
dew point stays slightly above the temperature during the entire time series of this flight, which indicates water-saturated to
water-supersaturated conditions. However, the trend in air speed measured by the pss8 around 13:00 UTC is artificial as the
tubing of the system got clogged by cloud water. Such events need to be identified manually as there is no sensor within the
system which could automatically detect water inside the tubing. The instrument orientation was very stable in pitch and roll
angle, but the yaw angle is characterised by sudden changes of more than 30° within a few seconds. Vertical wind shear and
convection may partly explain this trend, as the balloon might have been exposed to a different wind speed and direction than
the MPCK$^+$.



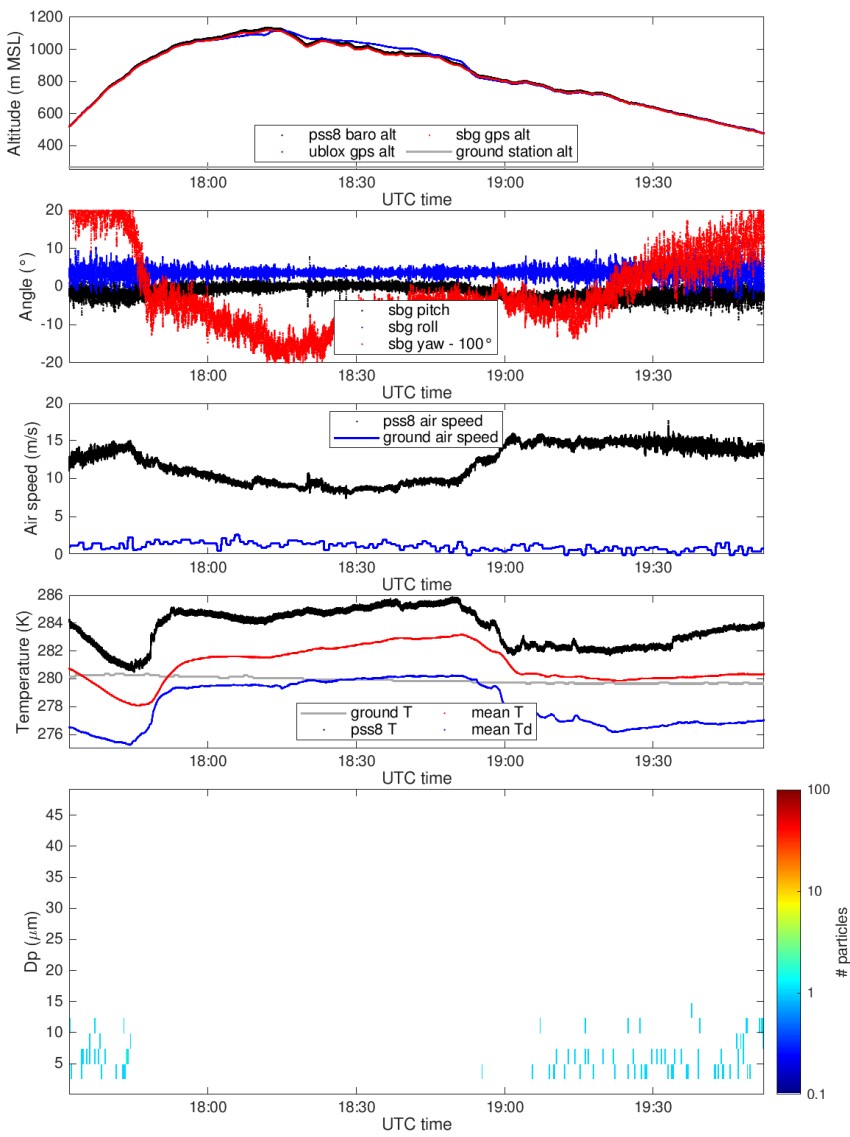

**Figure 6.** Time series data of altitude measurements (top panel), platform orientation (second panel), wind speed (third panel), temperature and dew point (fourth panel), and particle size and concentration measured by the FCDP (bottom panel) during Flight 20220921.1721. Instrument and quantity names in panels 1-4 are mentioned in each legend. All time data is presented in UTC, with local time being UTC+3 hours.

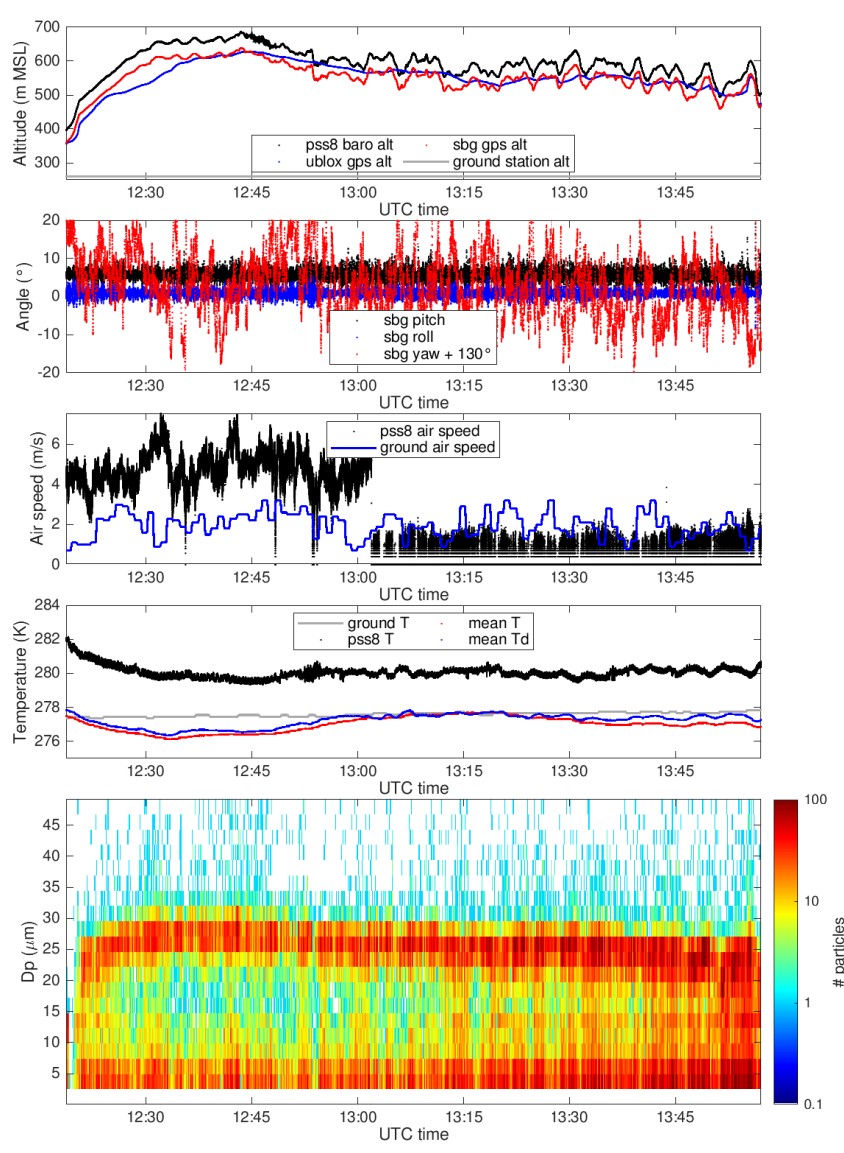

**Figure 7.** As in figure 6 for Flight 20220926.1209.



## 3.5 Temperature and humidity profiles

In addition to the time series data, the vertical profile data of temperature and dew point allow some conclusions about the meteorological situation during the measurements. For all eight flights, these profiles are summarised in figure 8. In the data of Flight 20220921.1721 we see an almost isothermal layer from the ground to 500 m altitude with an inversion layer between 500 and 600 m altitude. While the FCDP data in figure 6 indicate few particles below 800 m MSL (which is about 500 m above ground), there are no particles detected at higher altitude. So we can conclude that the aerosol particles in the boundary layer did not reach the free atmosphere during this measurement. Two more contrasting examples are Flight 20220923.0914 and Flight 20220925.0839 as the temperature profile decreases with altitude and the dew point data indicate water-saturated conditions at some altitude. During these two days, there was cumulus convection and a few clouds were sampled with the MPCK+. The trend in temperature and dew point during Flight 20220926.1209 is quite interesting as the data indicate a rapid cooling around 270 m above ground. It could be the top of the first cloud layer and the decrease in temperature might be caused by diabatic effects from precipitation falling out of higher cloud layers.

## 3.6 Examples of holography data

As already mentioned in the data description, the published data set does not contain holograms or processed holograms from the MPCK+holography unit. Nevertheless, we like to show an example of the particle data that can be derived from one single image. Details about the hologram processing and data extraction are presented in Thiede et al. (2025). Figure 9 displays the position of each droplet with colour-coded diameter in the top panel, and the logarithmically weighted droplet size distribution in the bottom panel. Each dot in the top panel represents one droplet. This allows a detailed look into the spatial distribution of droplets to examine droplet clustering, the presence of vortices or other phenomena which are not accessible by one-dimensional data from e.g. FCDP measurements. The droplet size distribution in the bottom panel has its maximum close to 20 μm and another local maximum around 40 μm, which is in agreement with the droplet size distribution measured by the FCDP (compare the red and the blue curve in figure 9). The primary mode around 5 μm droplet diameter in the FCDP data is too small for the holography unit to be measured, as its effective pixel pitch is 3 μm and objects smaller than 2 pixels in diameter are not distinguishable from background noise. As found in previous studies, the droplet concentration measured with the holography unit is in the same range as the FCDP data for particles with at least 11 μm diameter. On the other hand, we see a statistical effect when comparing the particle concentration around 40 μm diameter: There was 1 droplet in the holographic sample volume of 8.6 cm$^3$ within this size bin whereas the FCDP detected 2 droplets in a volume of less than 1.5 cm$^3$. The size distribution statistics from the holography unit are more robust compared to the FCDP data, thanks to the sample volume which is about 6 times larger. The effect of counting statistics is also visible in the shaded area and in the FCDP median concentration in the bottom panel of figure 9.



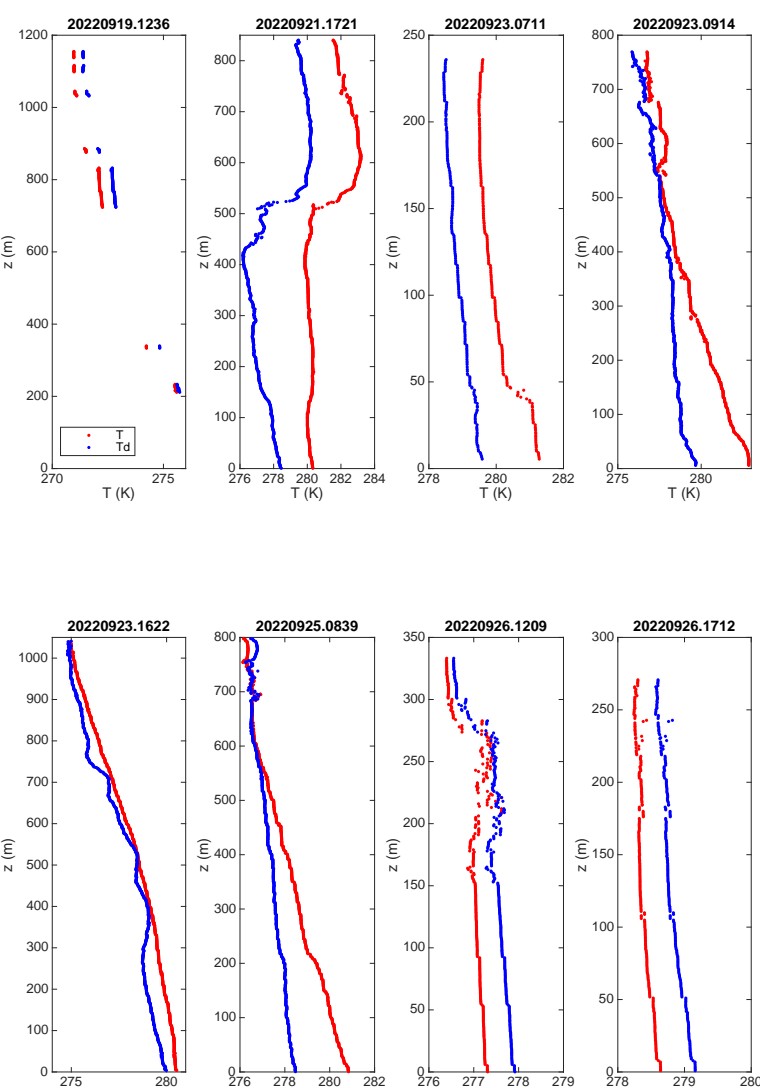

**Figure 8.** Vertical profiles of temperature and dew point for all eight MPCK⁺flights. Shown are mean temperature (red) and mean dew point (blue) as defined in Section 3.2 as a function of GPS altitude above ground. The flight names are given in the title of each panel.

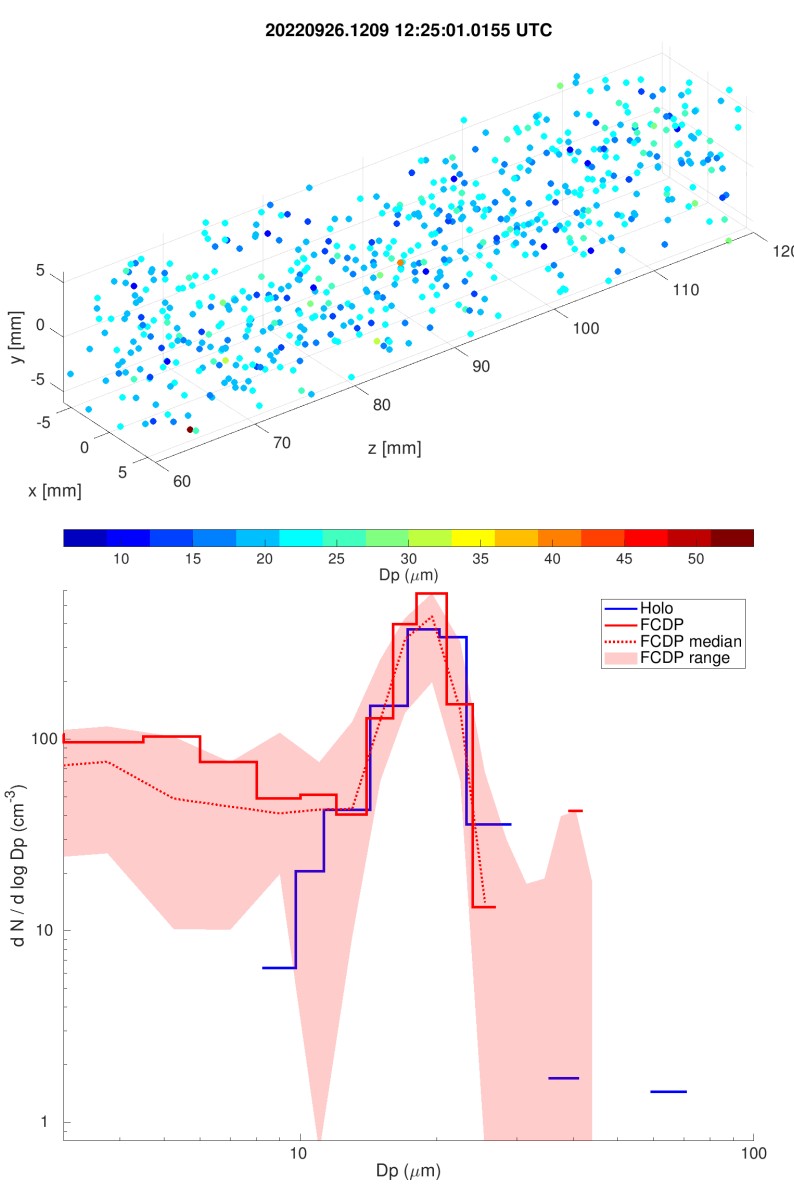

**Figure 9.** Example data from one hologram (number 86 in the series of Flight 20220926.1209) with particle position and colour-coded particle diameter in the top panel, and particle size distribution (dN/dlogDp) in the bottom panel. The blue curve represents the holography data, the red curve represents the FCDP data from the same second as the hologram was taken. The red shaded area reflects minimum and maximum droplet concentration measured by the FCDP from 10 s before until 10 s after the time the hologram was taken. The dotted red curve represents the median concentration of the entire 20 s period of FCDP data. Please keep in mind that the holography data are not included in the netCDF data linked to this paper.





### 3.7 Platform orientation

The orientation of the MPCK$^+$measured with the primary (sbg) and backup (powerinterlock) INS is summarised in figure 10 and 11 and table 5. There is an offset in pitch angle between sbg and powerinterlock of 10.4 °. Flight 10220919.1236 was excluded from the analysis as the repeated power outages precluded a GNSS fix of both INS. The median pitch angle measured by the sbg device was between -1.6 and +5.9 degrees with the higher values in the last three flights. However, we see the exact opposite trend in the roll angle statistics with higher angles in the first five flights. The agreement between sbg and powerinterlock is better than 1 °for the roll angle. The highest variability expressed by the standard deviation is seen in the yaw angle statistics. As the yaw angle is primarily determined by the course of the Helikite, wind shear in the boundary layer is one of the reasons for the high variability. The histograms in figure 10 show a relatively narrow angular distribution with a full width at half maximum (FWHM) of 3-6 °with a maximum close to zero. We conclude that the orientation of the MPCK$^+$was stable and nearly horizontal. However, the yaw angle has a much wider range as seen in figure 11. This is mainly due to the fact that wind shear has a strong influence on the position of the Helikites, which determines the course of the MPCK$^+$.

### 3.8 Technical Validation

#### 3.8.1 Sensor calibration

The hmp7 sensor was factory calibrated before the field campaign. The rht sensor was temperature calibrated in-house against a calibrated NTC (which was calibrated against a NIST-traceable LakeShore Pt-100 reference thermistor with a calibration uncertainty of $\pm$ 20 mK). Humidity calibration was done by placing the sensor in a sealed chamber above a saturated brine with both air temperature and solution temperature logged. The humidity fixed points of different salts were taken from Greenspan (1977). We used lithium chloride, potassium acetate, magnesium chloride, potassium carbonate, sodium bromide, sodium chloride, potassium chloride and potassium sulfate in a temperature range between 20 and 25 °C. In addition, we calibrated the rht sensor against water saturation (sensor was placed above a reservoir of distilled water) and against zero relative humidity (air was replaced by pure dry nitrogen). It should be noted that conditions during a balloon flight are different from conditions in an air-conditioned laboratory environment. Due to that, we found deviations from the idealised test conditions in particular in the relative humidity measurements (hmp7 reported too high relative humidity whereas rht reported too low relative humidity). Also, the sensors are not perfectly protected against rain, which can lead to very high relative humidity values even under water-subsaturated conditions.

#### 3.8.2 Plausibility checking and processing

During the Level 1 data processing scheme, the different instrument clocks are synchronised. In addition, the data are parsed and checked for sanity. Missing data or implausible values (e.g. static air pressure below 700 hPa despite a maximum tether length of 2 km) receive a flag to mark these data as not consistent. A quality flag value of 0 indicates valid data.





| Flight | Device | Quantity (unit) | Median | STD | Min | Max |
|---|---|---|---|---|---|---|
| 20220921.1721 | sbg | pitch angle (°) | -1.2 | 1.6 | -9.4 | 7.8 |
| 20220921.1721 | sbg | roll angle (°) | 3.7 | 1.8 | -10.1 | 16.7 |
| 20220921.1721 | sbg | yaw angle (°) | 97.1 | 12.4 | 77.0 | 160.0 |
| 20220921.1721 | powerinterlock | pitch angle (°) | -11.7 | 1.6 | -19.4 | -2.9 |
| 20220921.1721 | powerinterlock | roll angle (°) | 2.9 | 1.8 | -10.5 | 14.9 |
| 20220923.0711 | sbg | pitch angle (°) | 1.5 | 2.0 | -9.5 | 13.8 |
| 20220923.0711 | sbg | roll angle (°) | 3.6 | 0.8 | -1.4 | 8.4 |
| 20220923.0711 | sbg | yaw angle (°) | 165.7 | 22.8 | 111.8 | 289.9 |
| 20220923.0711 | powerinterlock | pitch angle (°) | -8.8 | 1.9 | -19.2 | 2.6 |
| 20220923.0711 | powerinterlock | roll angle (°) | 3.5 | 0.8 | -1.3 | 8.3 |
| 20220923.0914 | sbg | pitch angle (°) | -0.1 | 1.3 | -9.6 | 9.6 |
| 20220923.0914 | sbg | roll angle (°) | 4.2 | 0.7 | 0.5 | 7.5 |
| 20220923.0914 | sbg | yaw angle (°) | 153.3 | 9.3 | 96.2 | 240.7 |
| 20220923.0914 | powerinterlock | pitch angle (°) | -10.5 | 1.3 | -18.9 | -0.9 |
| 20220923.0914 | powerinterlock | roll angle (°) | 4.1 | 0.7 | 0.4 | 6.9 |
| 20220923.1622 | sbg | pitch angle (°) | -1.6 | 2.0 | -9.7 | 9.6 |
| 20220923.1622 | sbg | roll angle (°) | 4.4 | 1.6 | -6.2 | 14.7 |
| 20220923.1622 | sbg | yaw angle (°) | 160.2 | 7.4 | 145.4 | 239.7 |
| 20220923.1622 | powerinterlock | pitch angle (°) | -12.5 | 1.9 | -19.8 | -1.4 |
| 20220923.1622 | powerinterlock | roll angle (°) | 4.2 | 1.6 | -5.3 | 14.1 |
| 20220925.0839 | sbg | pitch angle (°) | 4.6 | 1.8 | -8.6 | 17.7 |
| 20220925.0839 | sbg | roll angle (°) | 3.1 | 2.5 | -26.4 | 19.0 |
| 20220925.0839 | sbg | yaw angle (°) | 254.4 | 22.7 | 167.9 | 254.4 |
| 20220925.0839 | powerinterlock | pitch angle (°) | -6.6 | 1.7 | -18.2 | 5.8 |
| 20220925.0839 | powerinterlock | roll angle (°) | 3.0 | 2.4 | -20.0 | 16.7 |
| 20220926.1209 | sbg | pitch angle (°) | 5.6 | 1.6 | -4.5 | 21.5 |
| 20220926.1209 | sbg | roll angle (°) | 0.9 | 1.0 | -8.4 | 7.2 |
| 20220926.1209 | sbg | yaw angle (°) | -127.0 | 9.0 | -160.1 | -87.4 |
| 20220926.1209 | powerinterlock | pitch angle (°) | -5.4 | 1.5 | -14.2 | 6.1 |
| 20220926.1209 | powerinterlock | roll angle (°) | 0.8 | 0.9 | -7.0 | 5.4 |
| 20220926.1712 | sbg | pitch angle (°) | 5.9 | 1.4 | -6.8 | 17.6 |
| 20220926.1712 | sbg | roll angle (°) | 1.6 | 0.5 | -2.2 | 6.3 |
| 20220926.1712 | sbg | yaw angle (°) | -141.5 | 10.9 | -163.7 | -100.5 |
| 20220926.1712 | powerinterlock | pitch angle (°) | -5.4 | 1.3 | -19.1 | 4.0 |
| 20220926.1712 | powerinterlock | roll angle (°) | 1.3 | 0.5 | -1.7 | 5.2 |

**Table 5.** Statistical summary of the measured MPCK[+]platform orientation expressed in Euler angles. Displayed are median, standard deviation, minimum and maximum of each angle from the respective device for 7 of the 8 flights. Data for Flight 20220919.1236 are not shown due to issues getting a GNSS fix.

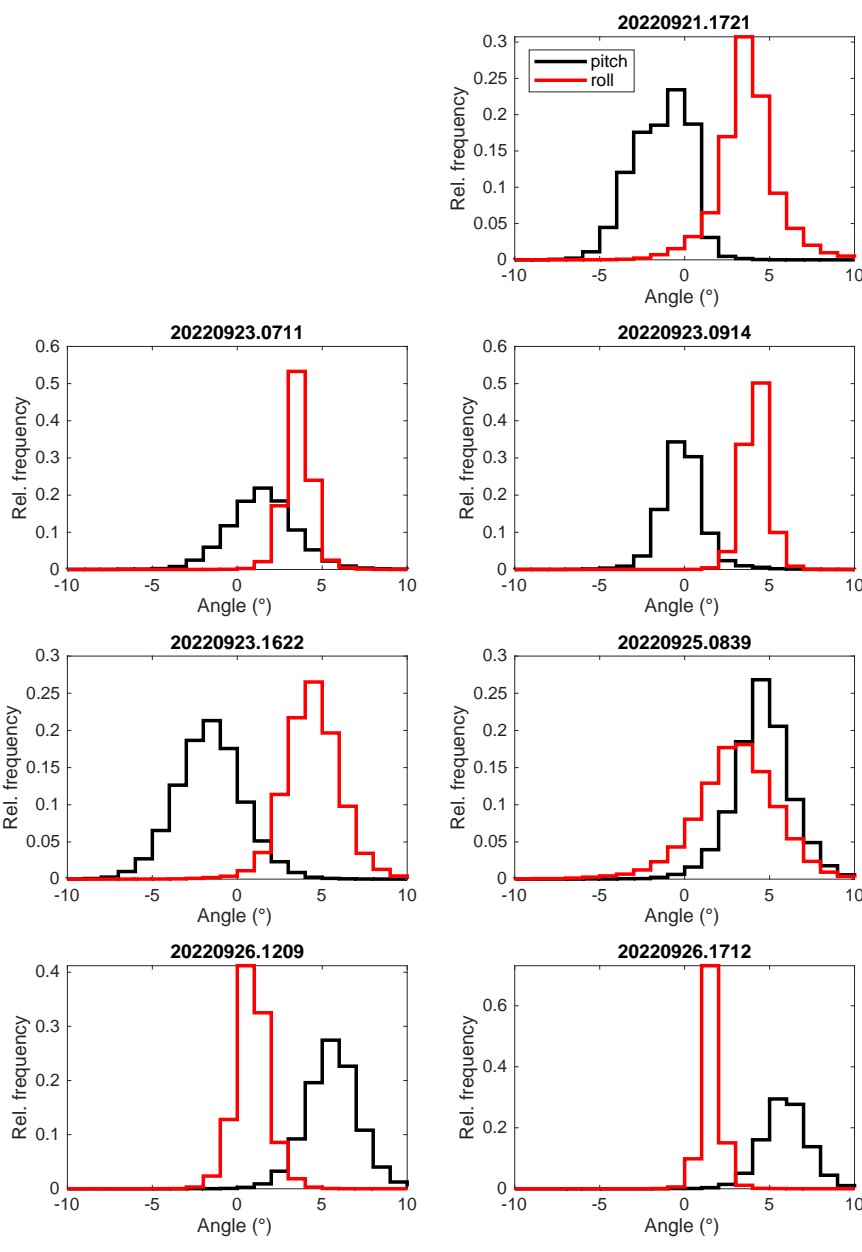

**Figure 10.** Histograms of pitch and roll angle for seven of the eight flights with MPCK⁺. Data were taken for those times where the MPCK⁺ was 50 m or higher above the ground.

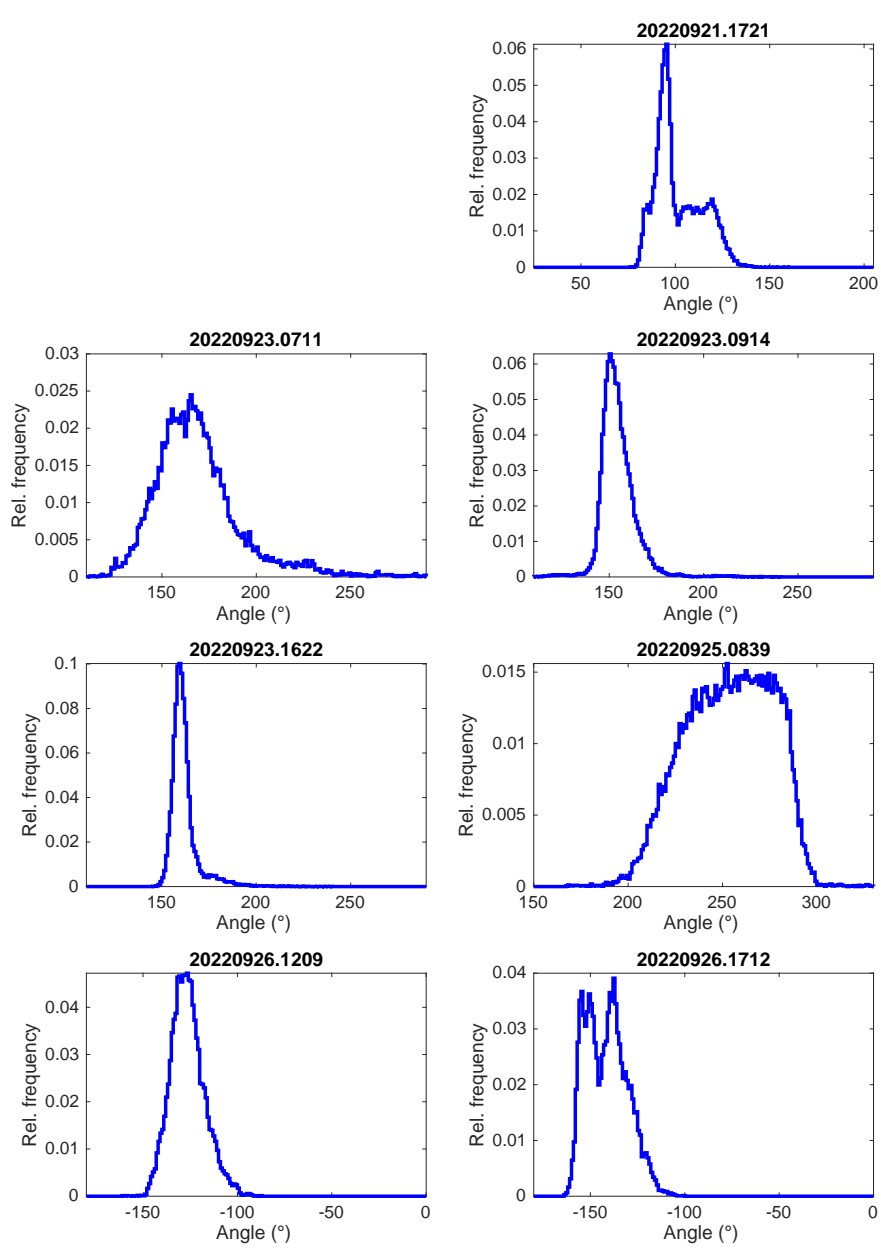

**Figure 11.** As in figure 10 for the yaw angle.





*Data availability.* All files are archived under individual DOIs at the Zenodo Open Science data archive (zenodo.org) where a dedicated community, Pallas Cloud Experiment – PaCE2022, has been established. The data files are available under the following URL: https://doi.org/10.5281/zenodo.14938135 (Bagheri et al., 2025b). This community hosts the data files along with additional metadata related to the data sets. Source code (in Python) for reading the netCDF files and plotting some of the core variables is published along with the data files.

## 4   Usage Notes and possible end users

The data set derived from the MPCK$^+$instrument platform is intended for atmospheric scientists who study turbulence in the atmospheric boundary layer and cloud-turbulence interaction. To get the full picture of boundary layer dynamics and thermodynamics, we recommend combining this data set with the WinDarts and ground weather station data sets published in Chavez-Medina et al. (2025).

The data set provides

– Example cases of clouds in the lower atmospheric boundary layer

– Inputs for model development

– Synergies with complementary measurements during PaCE 2022, including remote sensing data, UAV observations, WinDart data and ground weather station data

As already stated in Section 3.2, there are some known issues of the temperature and humidity calibration. First of all, the 300   pss8 temperature sensor is positively biased w.r.t. the mean temperature computed from the hmp7 and rht sensor. Moreover, the humidity of the hmp7 with the factory calibration applied is too high. We recommend our algorithm of taking the arithmetic mean from the HMP7 and RHT temperature, relative humidity and dew point data to calculate the most accurate temperature, relative humidity and dew point.

Besides the averaging of temperature and humidity data, the pitch angle measured by the ublox GNSS of the powerinterlock 305   device needs to be corrected by adding an offset correction of 10.4 °to the data.

Following the discussion in Section 3.1, corrections must be applied to the measured air speed in order to obtain the undisturbed air speed, i.e., the air speed far from the MPCK$^+$. Additional corrections are also necessary for computing the FCDP's swept volume, which relies on a further refined air speed. Moreover, the air speed measured by the pss8 and the particle counts recorded by the FCDP differ in both sampling frequency and timestamps, and each dataset includes its own validity flags 310   and missing values. Consequently, users must determine how best to interpolate and filter the air speed data when computing droplet concentration. To assist with this process, we provide a Python code sample along with the data that can serve as a starting point. This code applies the required corrections and generates essential variables, including droplet concentration, for those who wish to work directly with the supplied data.



*Author contributions.* GB, FN and OS designed and assembled the instrument. Mechanical parts and electronics were developed in collaboration with and manufactured by the in-house machine shop and scientific electronics teams. FN, OS, CEB, and EB performed in-situ measurements and collected the data. GB, FN and OS developed the software for data acquisition and data analysis. OS performed the data analysis, prepared the figures and wrote the first draft of this manuscript. BT and OS processed the holograms and analysed the holographic data. FN did the level 1 data processing and VCM reprocessed the netCDF files for this publication. All authors contributed to writing the final version of the manuscript.

*Competing interests.* The authors declare no competing interests. The sponsors had no influence on the study design, data collection and analysis, decision to publish, or preparation of the manuscript.

*Acknowledgements.* We would like to thank David Brus from the Finnish Meteorological Institute for making it possible for us to take part in the campaign and for his active support during our field work. We thank the staff of MPI-DS machine shop and research electronics group for their support in preparing the components of the MPCK⁺instrument platform, the ground anchors and additional parts necessary for flying the MPCK⁺. Furthermore, we like to acknowledge the support by the MPI-DS High Performance Computation (HPC) team in configuring and installing the HPC unit inside the Mobile Cloud Laboratory vehicle. Thanks go to Andreas Kopp and Dr. Artur Kubitzek for organising the campaign logistics, and to Constantin Schettler and Marcel Meyer for operating the MPCK and supporting the scientists at the launch site. Development of the MPCK platform and the MPCK⁺instrument was funded by internal funds of the Max Planck Society.



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
