# Peer review of "Airborne measurements of turbulence and cloud microphysics during PaCE 2022 using the Advanced Max Planck CloudKite Instrument (MPCK+)"

_Earth System Science Data, 2025_

## Referee Comment (RC1)

Review about:

**Airborne measurements of turbulence and cloud microphysics during PaCE 2022 using the Advanced Max Planck CloudKite Instrument (MPCK+)**

By
Oliver Schlenczek, Freja Nordsiek, Claudia E. Brunner, Venecia Ch.vez-Medina, Birte Thiede, Eberhard Bodenschatz, and Gholamhossein Bagheri

First of all, I would like to congratulate the authors on these measurements; I can judge from my own experience how complicated measurements of this kind are. The setup on which these measurements are based is quite impressive.

Nevertheless, I have a few critical general comments first. I would like to divide the presented measurements into two parts: standard meteorological parameters (wind, pressure, temperature and humidity) and, as a second set, perhaps the real challenges such as cloud droplets and high-resolution turbulence which makes the observations definitively unique. The first area is discussed here and data is presented although I have to say that the discussion of the temperature and humidity data in particular raises a few questions, which I have specified below.

The second part is mentioned but taken out of the published data set (holography, PVT, hotwire). This makes the manuscript seem somehow off to me and raises questions. Why are the comparatively simple measurements that every radiosonde can perform discussed in detail, and why are the really elaborate and complicated measurements – which are highly appreciated -  mentioned but not published?

Furthermore, I would also expect a quantitative assessment of the data quality which is missing.

The accuracy of the two inertial systems has been discussed in some detail, although I don't really understand the large offset in pitch. What I am missing, however, is an explanation of which measurements really require the Euler angles (roll & pitch) with the corresponding accuracy, as long as a three-dimensional wind vector is not to be determined – which is not planned with a one-component hotwire or Pitot tube.

I think this manuscript needs some more work before it can be considered for publication.

More specific:

Abstract:

Just out of curiosity: why is 'Advanced' capitalized in the full acronym even though the 'A' does not appear in the acronym?

Line 4: I suggest to include "cloudy" before planetary boundary layer to make clear that you sampled boundary layer clouds

Line 6: Well, Pallas is at 68°N and if you consider locations North of the Polar circle as "Arctic" you are right but I suggest to say "Polar regions" or so but this is a personal opinion.

Line 8: the mentioned distance between two cloud droplet size distributions is somewhat misleading; the sampling time and statistical significance is quite complex. Obviously, you store a size distribution every one second and assume less than 10 m/s true airspeed (wind speed)?! Is this a limitation of your system or do you sample individual droplets an estimate a size distribution over one second (later on this is discussed in some detail)? How robust is this estimate? Please clarify!

Line 9ff: To say that aircraft cannot fly in clouds at such low altitude is somewhat to hard. I suggest to soften this statement a little bit, look at the Polar research aircraft of AWI – they fly quite low (over open water) even in clouds and partly also in super-cooled clouds. Maybe you should argue that you can do observations even of low-level clouds/fog where it is often too dangerous for aircraft.

Introduction:

Line 17: Although I know what you are mean: this is only true for a fixed sampling rate – right? In this context, I would also mention that many problems such as adiabatic heating of inlets (consider temperature measurements from fast flying aircraft) and hazards like droplet shattering are minor if using a tethered balloon.

Line 33: is there a reason why not including the imaging data into this data set. It would make the data set much more complete!

Line 38: Please provide details where the spatial scale of less 10 m for size distributions comes from, this statement is somewhat vague (see also my comment about the abstract)

Line 67: Please provide a few more details/arguments why the helikite has an advantage compared to a classical tethered balloon. From your explanation it is not clear for most of the readers (although I know both systems).

Line 71: Please provide one sentence about the advantage to combine two helikites. Is it more stable or just the higher lift?

Line 72/73: I don't really understand this argument.

Line 82: Do you really think that the nonlinearity between tether length and altitude is an issue? If you have different layers with different wind speeds (for example a low-level jet) you have always the "problem" that you rope is not straight. You measure the height above ground – that should be sufficient – right?

About fig 2: from the description it is not clear for me why the right part of the main tether goes upward? Perhaps you could make a sketch that shows the entire line guide?

Line 112: the flight descriptions such as "20220919.1236" should be explained although one could imagine what it means. Also is the time local or in UTC? Later on, you explain that all times are in UTC but I think a short comment at this point would help.

Line 116ff: Although I like this kind of description of the individual weather conditions during the individual flights, it is a bit confusing and difficult to read. Why not a small paragraph for each of the flights (there aren't too many) instead of a continuous text? Maybe a separated into meteorological conditions and technical issues?

Line 139: the last argument of the description of "b1" is misleading and maybe at the end it includes a typo? Please consider re-wording.

Line 145: you already explained the acronyms in tab 1 so this is somewhat repetitive.

Line 152: tab 1 or 2?

Line 153/4: The editor must decide whether it is common practice to keep parts of the data set under lock and key; in my view, at least a justification should be provided for why this is the case. The fact that the data can be obtained from the authors if there is a 'justified' interest is somewhat confusing for a data paper.

Line 174: I am confused by this nomenclature/numbers

Section 3.3: I am a little bit confused here. You didn't provide any technical information about the sensors itself. However, the Vaisala device for example is based on the classical (heated) humicap – a capacitive sensor measuring the relative humidity with quite high absolute accuracy of 0.8% (as stated in the manual) but only up to 90% RH. So why not make use of the clouds assuming a mean RH = 100%. If you trust the temperature measurements you could calculate the dew point temperature and compare with the readings of the HMP7. I think this would be a more convincing way to describe the accuracy of this device.

Furthermore, I am not quite sure whether I have understood the statements regarding the accuracy of the temperature and humidity sensors correctly. Is it correct that you only get plausible humidity values if you take the arithmetic mean of two different sensors? If that is really the intention, it does not convince me.

For me, the discussion in Sec 3.3 is somewhat vague and I don't get any feeling about the accuracy of the observations. Even more, I would like to get some numbers describing the accuracy.

Section 3.4.: Does the use of a dual band GNSS really explain the GPS-derived height deviation between the two devices of 10 to 20 m or so? Have you compared to barometric height which should be of high accuracy?

Line 202: I don't quite follow this argument about the temperature below 200 m – especially Fig. 6 don't show data below 500 m ASL – right? And why should a discrepancy with the

ground temperature support any statement here? Can a ground-level inversion be completely ruled out? Is there a model for the response time of the sensors? If so, shouldn't it be a simple matter to assume a first-order system and correct for this response time?

I think the time series of a ground-level temperature doesn't really help to understand profile measurements, or have I misunderstood your argument?

At least for the sections shown in Fig. 6, the wind speed is also quite high. If temperature sensors cannot adapt quickly to the environment under these conditions, the design could be a problem, couldn't it?

Everyone is familiar with the problem at low flow velocities, but that is not the case here...

line 213ff: I think there is nothing wrong with the pure description of the cloud drop distribution, but after the formulation it sounds a bit like it would be a continuous process: activation at 5 microns and drop growth at a diameter of 25 microns - but what happens in between? It could almost be a secondary activation - right?

Sec 3.5

The first thing I notice in the profiles are the dew point temperatures, which are above the current temperature in the last two profiles. That is physically questionable – have the colors been mixed up?

Regarding the second last profile and the temperature decrease at the cloud top: without it being explicitly mentioned in the text, I suspect that it is about ascents? How is it prevented that after the cloud passage the sensors have become wet and the drops evaporate when exiting the cloud and artificially cool the sensor?
This would probably only cause a rather short cooling - but should be mentioned as a known problem. But if, as you have speculated, it is a case of evaporating cloud droplets from higher cloud layers, why is the dew point parallel to the temperature? Without being able to prove it with the data: I suspect that it is advection of a cooler layer, but that is just as speculative.

Section 3.7: An inclination deviation of about 10° is remarkable – does anyone have any idea what the cause of this could be? With such a compact system, installation errors should actually be smaller – are there any laboratory tests that can indicate which sensor is correct?

About the yaw angle: As I understand, the yaw angle is the platform heading e.g., the orientation around the vertical axis in an Earth-fixed system. So why is it depending of the course of the helikite? It should be – if the tail makes its job – depending on the wind direction – right? Saying this, I am not totally convinced if I can learn something from Fig 11. Fig 10 tells me something about the system performance and I agree that the payload seems to be quite stable.

---

## Author Comment (AC1)

**Reply to Referee 1**

July 11, 2025

Dear Referee,

Thank you very much for reviewing our manuscript. We are very grateful for the extremely helpful and constructive comments. In the following, we provide point-by-point replies to the points raised in your report. We have written the original text of the reviews in black colour and our response in red colour.

**General comments:**

First of all, I would like to congratulate the authors on these measurements; I can judge from my own experience how complicated measurements of this kind are. The setup on which these measurements are based is quite impressive. Nevertheless, I have a few critical general comments first. I would like to divide the presented measurements into two parts: standard meteorological parameters (wind, pressure, temperature and humidity) and, as a second set, perhaps the real challenges such as cloud droplets and high-resolution turbulence which makes the observations definitively unique. The first area is discussed here and data is presented although I have to say that the discussion of the temperature and humidity data in particular raises a few questions, which I have specified below. The second part is mentioned but taken out of the published data set (holography, PVT, hotwire). This makes the manuscript seem somehow off to me and raises questions. Why are the comparatively simple measurements that every radiosonde can perform discussed in detail, and why are the really elaborate and complicated measurements – which are highly appreciated - mentioned but not published?

Thank you for your comments. The main reason for not publishing the PIV and holography data along with the standard data is the sheer size of these data. PIV images are on the order of 250 GB and holograms take about 8 TB of disk space. The second reason is the fact that only a very small piece (less than 0.1 %) of the whole data set is already processed, so most of these data are just raw images, which are not very helpful for the user without special knowledge and software to process them. It is planned to publish these data at a later time when the processing and quality control are finished, but this is impossible before the Special Issue for PaCE 2022 is closed.

Furthermore, I would also expect a quantitative assessment of the data quality which is missing.

We have provided more details about the data quality in the respective sections.

The accuracy of the two inertial systems has been discussed in some detail, although I don't really understand the large offset in pitch. What I am missing, however, is an explanation of which measurements really require the Euler angles (roll& pitch) with the corresponding accuracy, as long as a three-dimensional wind vector is not to be determined – which is not planned with a one-component hotwire or Pitot tube.

Originally, the instrument was equipped with a 3D Pitot tube, which was not available for this campaign. The offset is just due to a typo in the configuration settings and is corrected in the next release of the data.

I think this manuscript needs some more work before it can be considered for publication.

**More specific:**

Abstract:

Just out of curiosity: why is 'Advanced' capitalized in the full acronym even though the 'A' does not appear in the acronym?

This is for historical reasons and goes back to previous papers we have published. We realise that the name of the instrument box is not ideal and is very similar to the name of the platform, but changing it at this stage would only cause more confusion. We have added some details about the instrument name that will hopefully make it clearer why we have the word "Advanced" in there.

Line 4: I suggest to include "cloudy" before planetary boundary layer to make clear that you sampled boundary layer clouds

Thank you for the suggestion, we added "cloudy".

Line 6: Well, Pallas is at 68°N and if you consider locations North of the Polar circle as "Arctic" you are right but I suggest to say "Polar regions" or so but this is a personal opinion.

Other authors of manuscripts submitted to this ESSD Special Issue used the term "sub-arctic", so we decided to change our text accordingly.

Line 8: the mentioned distance between two cloud droplet size distributions is somewhat misleading; the sampling time and statistical significance is quite complex. Obviously, you store a size distribution every one second and assume less than 10 m/s true airspeed (wind speed)?! Is this a limitation of your system or do you sample individual droplets and estimate a size distribution over one second (later on this is discussed in some detail)? How robust is this estimate? Please clarify!

We mentioned the typical spatial separation based on one-second FCDP (Fast Cloud Droplet Probe) data and a mean wind of 10 m/s. Using particle-by-particle (pbp) data would allow for smaller spatial separation but with much higher statistical uncertainty due to low counting statistics. However, one can use the published pbp data to calculate parameters of interest with a spatio-temporal resolution other than 1 Hz. No changes to the text have been made.

Line 9ff: To say that aircraft cannot fly in clouds at such low altitude is somewhat to hard. I suggest to soften this statement a little bit, look at the Polar research aircraft of AWI – they fly quite low (over open water) even in clouds and partly also in super-cooled clouds. Maybe you should argue that you can do observations even of low-level clouds/fog where it is often too dangerous for aircraft.

Thank you, we agree that the wording was chosen too strong here. We removed "not accessible by research airplanes".

Introduction: Line 17: Although I know what you are mean: this is only true for a fixed sampling rate – right? In this context, I would also mention that many problems such as adiabatic heating of inlets (consider temperature measurements from fast flying aircraft) and hazards like droplet shattering are minor if using a tethered balloon.

Yes, we agree. We added the following text: "and avoids effects like adiabatic heating of inlets and shattering of droplets or ice crystals".

Line 33: is there a reason why not including the imaging data into this data set. It would make the data set much more complete!

There are two reasons for not adding the imaging data. One is the huge size of these data sets and the other is the fact that these data require very careful quality control and quality assurance, and only a few examples (much less than 0.1 % of the total) have been fully processed yet. So there are a few data gathered from the images which are of quality level b1, but the vast majority is still at a0 (raw data). Our intention is to share only the data of quality level b1. We modified the sentence in the following way and moved it to Section 2.3 as suggested by Referee 2: "Data acquired by the two imaging units are not part of the published data set due to ongoing QC/QA procedures, but the flight table provides information about availability of these data".

Line 38: Please provide details where the spatial scale of less 10 m for size distributions comes from, this statement is somewhat vague (see also my comment about the abstract)

The data are separated by one second in time, which yields 10 m spatial separation for a typical wind speed of 10 m/s. We added ", which depends on wind speed and acquisition rate". The text was moved to Section 2.3 as suggested by Referee 2.

Line 67: Please provide a few more details/arguments why the helikite has an advantage compared to a classical tethered balloon. From your explanation it is not clear for most of the readers (although I know both systems).

A classical tethered balloon with a circular shape is being pushed towards the ground in high wind conditions, which may cause damage to the scientific payload. A helikite stabilises in an angle around 45 degrees in windy conditions. We changed the sentence in the following way: "A Helikite is a balloon-kite combination whose main advantage over an ordinary balloon is the stable position in windy conditions. While ordinary balloons are being pushed towards the ground, a Helikite is stabilised by keel and sail underneath the bubble in a 45  $^{\circ}$  angle to the vertical."

Line 71: Please provide one sentence about the advantage to combine two helikites. Is it more stable or just the higher lift?

Actually, the combination of two helikites has both advantages. Takeoff and landing

can be performed under much more controlled and safer conditions, and there is a slight increase of lift (when using a 34 cubic meter helikite on top of a 250 cubic meter helikite, the increase in lift is about 10 %). We added the following sentence: "This tandem configuration increases the static lift by 10 % and due to the stabilising effect by the small Helikite, takeoff and landing of the big Helikite can be performed under much safer conditions, in particular in gusty winds."

Line 72/73: I don't really understand this argument.

We removed this sentence as it is not necessary.

Line 82: Do you really think that the nonlinearity between tether length and altitude is an issue? If you have different layers with different wind speeds (for example a lowlevel jet) you have always the "problem" that you rope is not straight. You measure the height above ground – that should be sufficient – right?

It is just a detail which we thought is worth sharing with the reader. Some people might think that the angle is always 45 degrees, but this is only true if the payload is attached very close to the centre of mass of the balloon. When planning airborne measurements with such a tethered balloon system, we think it is useful to know this potential limitation. Just thinking about some other research group wants to measure up to 2 km above ground and thinks it is sufficient to have 2.8 km of line. We did not modify this sentence.

About fig 2: from the description it is not clear for me why the right part of the main tether goes upward? Perhaps you could make a sketch that shows the entire line guide?

Thanks for the suggestion, we changed figure 2 to a sketch showing the whole setup from ground anchor to the top of the upper helikite.

Line 112: the flight descriptions such as "20220919.1236" should be explained although one could imagine what it means. Also, is the time local or in UTC? Later on, you explain that all times are in UTC but I think a short comment at this point would help.

We added the following sentence before the first flight description appears: "The flight identifier is the date of departure, followed by the takeoff UTC time (hours and minutes)."

Line 116ff: Although I like this kind of description of the individual weather conditions during the individual flights, it is a bit confusing and difficult to read. Why not a small paragraph for each of the flights (there aren't too many) instead of a continuous text? Maybe a separated into meteorological conditions and technical issues?

This is a good point. We have put the flights into individual paragraphs to optimise readability.

Line 139: the last argument of the description of "b1" is misleading and maybe at the end it includes a typo? Please consider re-wording.

Correct, this is a typesetting error. It is fixed now.

Line 145: you already explained the acronyms in tab 1 so this is somewhat repetitive. We changed the sentence to: "The device names are summarised in table 1".

Line 152: tab 1 or 2? Line 153/4: The editor must decide whether it is common practice to keep parts of the data set under lock and key; in my view, at least a justification should be provided for why this is the case. The fact that the data can be obtained from the authors if there is a 'justified' interest is somewhat confusing for a data paper. We decided to provide more details about our decision not to publish the imaging data along with this manuscript. The main reasons are the huge file size of the data sets and the labour needed for quality control. We selected one example in figure 9 just to demonstrate the value of these data after rigorous quality control (only this little part is at level b1). If someone requests a specific part of the data, it will take a while (typically weeks to months) to perform the final QC/QA before the data can be shared. There are known measurement artefacts that have to be considered in the analysis, which is way too detailed to put it in a manual and let the user just do it. We modified the sentence in the following way: "The additional groups, which are excluded from the published data, are labjack, holo, piv and serialcameras due to ongoing QC/QA procedures."

Line 174: I am confused by this nomenclature/numbers

Again a typesetting issue, which is fixed now.

Section 3.3: I am a little bit confused here. You didn't provide any technical information about the sensors itself. However, the Vaisala device for example is based on the classical (heated) humicap – a capacitive sensor measuring the relative humidity with quite high absolute accuracy of 0.8% (as stated in the manual) but only up to 90% RH. So why not make use of the clouds assuming a mean RH = 100%. If you trust the temperature measurements you could calculate the dew point temperature and compare with the readings of the hmp7. I think this would be a more convincing way to describe the accuracy of this device.

We added another column to table 1 with the sensor accuracy and/or measurement range, depending on the sensor type. Regarding the accuracy of humidity measurements, we actually just know that RH should not be below 100 % if the instrument is inside a cloud. The possible supersaturation should in theory be low (1 or 2 % typically in clouds which are not strongly convective), but we actually do not know the supersaturation. There was an issue in the JSON file for processing the hmp7 data, so the processed data had a positive offset in relative humidity of 10 %. This offset has been corrected now, which leads to a much better agreement between the hmp7 and the AM2315 sensor.

Furthermore, I am not quite sure whether I have understood the statements regarding the accuracy of the temperature and humidity sensors correctly. Is it correct that you only get plausible humidity values if you take the arithmetic mean of two different sensors? If that is really the intention, it does not convince me.

We did a thorough analysis of the temperature and relative humidity data and by applying an offset correction as described in the previous response, we get a much closer agreement between the two sensors. Figures 4 and 5 were updated and we changed the text referring to those figures accordingly: "The hmp7 dew point and temperature data suggest high humidity during the entire period while the FCDP detected particles in a few short events during the flight. As the deviation between the rht and hmp7 units in temperature is typically smaller than 200 mK, we trust both sensors but prefer the rht temperature data over the hmp7 temperature data due to the fact that the rht sensor is not heated. However, the rht dewpoint is too low in those sections where the FCDP detected cloud droplets. So we prefer the hmp7 dewpoint over the rht dewpoint."

A different example is shown in figure 5 for Flight 20220926.1209. Here, the instrument was flying in clouds for most of the entire flight time. Again, we see little deviation in the temperature data between rht and hmp7. The strong deviation in the dew point temperature is about the same as in the previous example, and also here, the dewpoint from the hmp7 sensor is more trustworthy than the dewpoint data from the rht sensor. Based on the results shown in figures 4 and 5, we decided to show only rht temperature, hmp7 dewpoint and ground station temperature in the following graphs instead of plotting data for each individual sensor.

For me, the discussion in Sec 3.3 is somewhat vague and I don't get any feeling about the accuracy of the observations. Even more, I would like to get some numbers describing the accuracy.

Based on the sensor inter-comparison in figures 4 and 5, we estimate the uncertainty in dewpoint to 0.5 K. Via inversion of the saturation vapour pressure we computed the relative humidity that results from using the rht temperature data and hmp7 dewpoint data. The maximum supersaturation from the corrected relative humidity data is 6.3 %. Given the measurement uncertainty at high humidity of  $\pm 2.4\%$  for the hmp7 sensor, this value appears reasonable.

Section 3.4.: Does the use of a dual band GNSS really explain the GPS-derived height deviation between the two devices of 10 to 20 m or so? Have you compared to barometric height which should be of high accuracy?

Yes, the GNSS difference can actually explain the deviation between the red (sbg) and blue (ublox) curve. The red curve follows the black curve (barometric altitude) very closely. From 18:01 to 18:53 UTC, we find a root-mean-square difference of 20.6 m between the barometric altitude and the ublox GPS altitude whereas the root-meansquare difference between the barometric altitude and the sbg GPS altitude is just 12.6 m. The minimum and maximum deviations between barometric altitude and ublox GPS altitude are -45 m and +42 m, the standard deviation is 21.9 m. The difference between sbg GPS altitude and barometric altitude has a much smaller variance (minimum -18 m, maximum -6 m, standard deviation 1.6 m). An offset between barometric and GPS altitude is expected as the air data computer of the pss8 uses some assumptions to calculate the altitude from the absolute pressure. To our knowledge, it uses the measured pressure and temperature and assumes a constant vertical temperature gradient. We added the following sentences in the section Technical validation / Plausibility checking and processing: "The barometric altitude of the pss8 device was taken directly from the air data computer. Both pss8 and sbg altitude data show the exact same trend in the time series, but there is an altitude-dependent offset between them. As the flights were short compared to meteorologically caused changes in surface air pressure, the barometric altitude was corrected by applying a linear correction based on intercomparison with GPS altitude from the sbg device. This results in an absolute accuracy of  $\pm$  15 m."

Line 202: I don't quite follow this argument about the temperature below 200 m – especially Fig. 6 don't show data below 500 m ASL – right? And why should a discrepancy with the ground temperature support any statement here? Can a ground-level inversion be completely ruled out? Is there a model for the response time of the sensors? If so, shouldn't it be a simple matter to assume a first-order system and correct for this response time? I think the time series of a ground-level temperature doesn't really help to understand profile measurements, or have I misunderstood your

argument? At least for the sections shown in Fig. 6, the wind speed is also quite high. If temperature sensors cannot adapt quickly to the environment under these conditions, the design could be a problem, couldn't it? Everyone is familiar with the problem at low flow velocities, but that is not the case here...

This is a good point. There is actually an inversion layer, as shown in Figure 8, second left top panel. We decided to remove this sentence and the two following sentences. Our intention in adding the ground level temperature and wind speed to Figures 6 and 7 was to put the time series and profile data of temperature and dewpoint measured by the MPCK+ into context with the conditions close to the ground. Due to the data correction described in the previous comments, we updated Figure 6 and changed the text referring to Figure 6 accordingly.

line 213: I think there is nothing wrong with the pure description of the cloud drop distribution, but after the formulation it sounds a bit like it would be a continuous process: activation at 5 microns and drop growth at a diameter of 25 microns - but what happens in between? It could almost be a secondary activation - right?

We cannot rule out secondary activation here, this is true. We have now modified the text to explain the main findings without speculating the underlying processes.

Sec 3.5 The first thing I notice in the profiles are the dew point temperatures, which are above the current temperature in the last two profiles. That is physically questionable – have the colours been mixed up?

The colours are definitely not mixed up and the data record of the hmp7 sensor had maximum RH around 106 %. By applying the offset correction described earlier, we could reduce this effect. The reprocessed data have a much smaller difference between temperature and dewpoint for the last two flights, but still the dewpoint is slightly higher than the temperature (about 0.5 K). We have added the following to the main text: "However, for flights 20220926.1209 and 20220926.1712, the dew point temperature is about 0.5 K higher than the air temperature, which is unphysical. This discrepancy can be attributed to measurement uncertainties arising from the combination of two sensors used to derive and correct these quantities, leading to a systematic bias in these two flights, both conducted in near-saturated conditions."

Regarding the second last profile and the temperature decrease at the cloud top: without it being explicitly mentioned in the text, I suspect that it is about ascents? How is it prevented that after the cloud passage the sensors have become wet and the drops evaporate when exiting the cloud and artificially cool the sensor? This would probably only cause a rather short cooling - but should be mentioned as a known problem. But if, as you have speculated, it is a case of evaporating cloud droplets from higher cloud layers, why is the dew point parallel to the temperature? Without being able to prove it with the data: I suspect that it is advection of a cooler layer, but that is just as speculative.

We removed the sentence "It could be the top of the first cloud layer and the decrease in temperature might be caused by diabatic effects from precipitation falling out of higher cloud layers." as it is too speculative.

Section 3.7: An inclination deviation of about 10 °is remarkable – does anyone have any idea what the cause of this could be? With such a compact system, installation errors should actually be smaller – are there any laboratory tests that can indicate which sensor is correct?

As already mentioned in the beginning of our response, the 10.4 degrees offset between the two instruments is due to a wrong number in the configuration file. It has been fixed in the revised version. Table 5 was corrected, the text explaining table 5 was changed accordingly.

About the yaw angle: As I understand, the yaw angle is the platform heading e.g., the orientation around the vertical axis in an Earth-fixed system. So why is it depending of the course of the helikite? It should be – if the tail makes its job – depending on the wind direction – right? Saying this, I am not totally convinced if I can learn something from Fig 11. Fig 10 tells me something about the system performance and I agree that the payload seems to be quite stable.

We agree that the yaw angle figure is not that much helpful, so we decided to remove it. Also removed the following text: "However, the yaw angle has a much wider range as seen in figure 11. This is mainly due to the fact that wind shear has a strong influence on the position of the Helikites, which determines the course of the MPCK+."

**Other author comments and modifications**

Figures 4 and 5 were updated. The plotting style was changed to increase readability.

Figures 6 and 7 were updated by using the new temperature and relative humidity data.

We removed the text in line 301 - 305 and adapted the sentence before in the following way: Moreover, the hmp7 sensor is heated, which may lead to a positive bias in temperature. Due to that, we recommend using the temperature recordings of the rht sensor and the relative humidity and dewpoint data in the corrected\_rht data group.

---

## Author Comment (AC2)

**Reply to Referee 2**

**July 11, 2025**

Dear Referee,

Thank you very much for reviewing our manuscript. We are very grateful for the helpful and constructive comments. In the following, we provide point-by-point replies to the points raised in your report. We have written the original text of the reviews in black colour and our response in red colour.

**Summary:**

This paper introduces a valuable dataset collected using the Advanced Max Planck CloudKite instrument (MPCK+) during the Pallas Cloud Experiment (PaCE) in Finland, September 2022. The dataset comprises high-resolution airborne measurements of turbulence, wind shear, and cloud microphysics from a unique tethered Helikite platform operating within the Arctic boundary layer. Key strengths include the high spatial resolution achieved due to the platform's low airspeed and the long-duration, low-altitude flight capabilities compared to traditional aircraft. The manuscript effectively details the instrumentation, campaign, data structure, processing, and provides useful examples. The authors are commendably transparent about data characteristics and potential challenges. This dataset represents a significant contribution for researchers studying boundary layer dynamics and cloud processes. This manuscript presents a unique and valuable high-resolution atmospheric dataset collected using the innovative MPCK+ tethered balloon platform during the PaCE 2022 campaign. The data provides important insights into Arctic boundary layer conditions and cloud microphysics, complementing other measurement strategies. The authors have provided a thorough description of the instrument, campaign, and data, including transparent discussion of data quality aspects. Overall recommendation: Addressing the minor comments listed below will strengthen the paper significantly. Therefore, I recommend this paper for publication in Earth System Science Data after these minor revisions have been addressed.

**Minor comments:**

Line 21-22: The similar acronyms "MPCK" for the platform and "MPCK+" for the instrument package could be confusing. Consider clarifying the naming convention. Does the "+" specifically denote "Advanced"? Is there a non-advanced instrument package, and if so, does it share the platform's acronym?

In previous publications we have already used the acronym MPCK for the platform and also for the instrument box. We have also built another instrument called mini-MPCK and a distinction had to be made between these instruments and the platform. In hindsight, the names could have been better chosen, but changing the names now would only cause more confusion. We have provided further details and explanations of various instruments developed for the MPCK platform in the main text.

Line 26-41: The detailed technical descriptions of the PIV/Holography units and other sensors currently appear in the Introduction. Consider relocating this information to Section 2.3 ("Instrumentation: the MPCK+") for better structural flow, keeping the introduction focused on the overall goals and dataset overview.

Thank you for the suggestion. We moved the text between "inline holography unit" and the end of the Introduction to Section 2.3 and merged it with the existing text in Section 2.3 to avoid duplications. After "inline holography unit" in Line 24, we added two sentences to describe the structure of this paper: Details about the field campaign, the MPCK platform, the instrumentation and the measurement flights are presented in Section 2. The data structure with examples is described in Section 3. Section 4 is dedicated to provide information for usage of the data.

Figure 1 Caption: Please add context to the caption, for example, specifying it shows a test flight and its approximate location (e.g., "near MPI-DS").

We added the following text: "next to the institute location of MPI-DS".

Table 1: For the PIV/PTV and Inline Holography entries, please specify the effective particle size range they can resolve or their measurement resolution limits.

We added the size range of particles for the imaging units.

Table 2: The "Instrument" column appears redundant as all flights use the MPCK+. Consider removing it.

Agreed. We decided to remove this column as it does not provide useful information to the reader.

Line 141: The current flight ID format is yyyymmdd.hhmm. Consider providing a simpler, sequential flight identifier (e.g., PACE22_FLT01, PACE22_FLT02...) in the metadata or tables for easier referencing across datasets or publications.

We appreciate the suggestion and agree that a simplified sequential flight identifier can be helpful for referencing. However, for consistency across the campaign, we decided to adhere to the naming convention used by all instruments and teams—based on yyyymmdd.hhmm—since not all platforms were deployed in every flight. A purely sequential flight numbering system would have made data integration and cross-platform comparisons more cumbersome and potentially confusing. That said, we did include a flight number identifier in the metadata of the flights.

Line 305: Clarify explicitly whether the noted 10.4° pitch offset correction required

for the powerinterlock device data has already been applied in the distributed NetCDF files or if users must apply it themselves. Table 5 appears to present uncorrected data.

In the previous version of the data and manuscript it has not been corrected. It is already corrected in the revised version of the data and the manuscript. The sentence has been removed.

**Other author comments and modifications**

Figures 4 and 5 were updated. The plotting style was changed to increase readability. Figures 6 and 7 were updated by using the new temperature and relative humidity data.

We removed the text in line 301 - 305 and adapted the sentence before in the following way: Moreover, the hmp7 sensor is heated, which may lead to a positive bias in temperature. Due to that, we recommend using the temperature recordings of the rht sensor and the relative humidity and dewpoint data in the corrected_rht data group.